# SPFQ: A Stochastic Algorithm and Its Error Analysis for Neural Network Quantization

## Abstract

Quantization is a widely used compression method that effectively reduces redundancies in over-parameterized neural networks. However, existing quantization techniques for deep neural networks often lack a comprehensive error analysis due to the presence of non-convex loss functions and nonlinear activations. In this paper, we propose a fast stochastic algorithm for quantizing the weights of fully trained neural networks. Our approach leverages a greedy path-following mechanism in combination with a stochastic quantizer. Its computational complexity scales only linearly with the number of weights in the network, thereby enabling the efficient quantization of large networks. Importantly, we establish, for the first time, full-network error bounds, under an infinite alphabet condition and minimal assumptions on the weights and input data. As an application of this result, we prove that when quantizing a multi-layer network having Gaussian weights, the relative square quantization error exhibits a linear decay as the degree of over-parametrization increases. Furthermore, we demonstrate that it is possible to achieve error bounds equivalent to those obtained in the infinite alphabet case, using on the order of a mere $\log \log N$ bits per weight, where $N$ represents the largest number of neurons in a layer.

## 1 Introduction

Deep neural networks (DNNs) have shown impressive performance in a variety of areas including computer vision and natural language processing among many others. However, highly overparameterized DNNs require a significant amount of memory to store their associated weights, activations, and – during training – gradients. As a result, in recent years, there has been an interest in model compression techniques, including quantization, pruning, knowledge distillation, and low-rank decomposition (Neill, 2020; Deng et al., 2020; Cheng et al., 2017; Gholami et al., 2021; Guo, 2018). Neural network quantization, in particular, utilizes significantly fewer bits to represent the weights of DNNs. This substitution of original, say, 32-bit floating-point operations with more efficient low-bit operations has the potential to significantly reduce memory usage and accelerate inference time while maintaining minimal loss in accuracy. Quantization methods can be categorized into two classes (Krishnamoorthi, 2018): quantization-aware training and post-training quantization. Quantization-aware training substitutes floating-point weights with low-bit representations during the training process, while post-training quantization quantizes network weights only after the training is complete.

To achieve high-quality empirical results, quantization-aware training methods, such as those in (Choi et al., 2018; Cai et al., 2020; Wang et al., 2019; Courbariaux et al., 2015; Jacob et al., 2018; Zhang et al., 2018; Zhou et al., 2017), often require significant time for retraining and hyperparameter tuning using the entire training dataset. This can make them impractical for resource-constrained scenarios. Furthermore, it can be challenging to rigorously analyze the associated error bounds as quantization-aware training is an integer programming problem with a non-convex loss function, making it NP-hard in general. In contrast, post-training quantization algorithms, such as (Choukroun et al., 2019; Wang et al., 2020; Lybrand & Saab, 2021; Zhang et al., 2023; Hubara et al., 2020; Nagel et al., 2020; Zhao et al., 2019; Maly & Saab, 2023; Frantar et al., 2022), require only a small amount of training data, and recent research has made strides in obtaining quantization error bounds for some of these algorithms (Lybrand & Saab, 2021; Zhang et al., 2023; Maly & Saab, 2023) in the context of shallow networks.

In this paper, we focus on this type of network quantization and its theoretical analysis, proposing a fast stochastic quantization technique and obtaining theoretical guarantees on its performance, even in the context of deep networks.

## 1.1 RELATED WORK

In this section, we provide a summary of relevant prior results concerning a specific post-training quantization algorithm, which forms the basis of our present work. To make our discussion more precise, let $X \in \mathbb{R}^{m \times N_0}$ and $w \in \mathbb{R}^{N_0}$ represent the input data and a neuron in a single-layer network, respectively. Our objective is to find a mapping, also known as a *quantizer*, $\mathcal{Q} : \mathbb{R}^{N_0} \to \mathcal{A}^{N_0}$ such that $q = Q(w) \in \mathcal{A}^{N_0}$ minimizes $\|Xq - Xw\|_2$. Even in this simplified context, since $\mathcal{A}$ is a finite discrete set, this optimization problem is an integer program and therefore NP-hard in general. Nevertheless, if one can obtain good approximate solutions to this optimization problem, with theoretical error guarantees, then those guarantees can be combined with the fact that most neural network activation functions are Lipschitz, to obtain error bounds on entire (single) layers of a neural network.

Recently, Lybrand & Saab (2021) proposed and analyzed a greedy algorithm, called *greedy path following quantization* (GPFQ), to approximately solve the optimization problem outlined above. Their analysis was limited to the ternary alphabet $\mathcal{A} = \{0, \pm 1\}$ and a single-layer network with Gaussian random input data. Zhang et al. (2023) then extended GPFQ to more general input distributions and larger alphabets, and they introduced variations that promoted pruning of weights. Among other results, they proved that if the input data $X$ is either bounded or drawn from a mixture of Gaussians, then the relative square error of quantizing a generic neuron $w$ satisfies

$$\frac{\|Xw - Xq\|_2^2}{\|Xw\|_2^2} \lesssim \frac{m \log N_0}{N_0} \tag{1}$$

with high probability. Extensive numerical experiments in (Zhang et al., 2023) also demonstrated that GPFQ, with 4 or 5 bit alphabets, can achieve less than $1\%$ loss in Top-1 and Top-5 accuracy on common neural network architectures. Subsequently, (Maly & Saab, 2023) introduced a different algorithm that involves a deterministic preprocessing step on $w$ that allows quantizing DNNs via *memoryless scalar quantization* (MSQ) while preserving the same error bound in 1. This algorithm is more computationally intensive than those of (Lybrand & Saab, 2021; Zhang et al., 2023) but does not require hyper-parameter tuning for selecting the alphabet step-size.

## 1.2 CONTRIBUTIONS AND ORGANIZATION

In spite of recent progress in developing computationally efficient algorithms with rigorous theoretical guarantees, all technical proofs in (Lybrand & Saab, 2021; Zhang et al., 2023; Maly & Saab, 2023) only apply for a single-layer of a neural network with certain assumed input distributions. This limitation naturally comes from the fact that a random input distribution and a deterministic quantizer lead to activations (i.e., outputs of intermediate layers) with dependencies, whose distribution is usually intractable after passing through multiple layers and nonlinearities.

To overcome this main obstacle to obtaining theoretical guarantees for multiple layer neural networks, in Section 2, we propose a new stochastic quantization framework, called stochastic path following quantization (SPFQ), which introduces randomness into the quantizer. We show that SPFQ admits an interpretation as a two-phase algorithm consisting of a data-alignment phase and a quantization phase. This allows us to propose two variants, both summarized in Algorithm 1, which involve different data alignment strategies that are amenable to analysis.

Importantly, our algorithms are fast. For example, SPFQ with approximate data alignment has a computational complexity that only scales linearly in the number of parameters of the neural network. This stands in sharp contrast with quantization algorithms that require solving optimization problems, generally resulting in polynomial complexity in the number of parameters.

In Section 3, we present the first error bounds for quantizing an entire $L$-layer neural network $\Phi$, under an infinite alphabet condition and minimal assumptions on the weights and input data $X$. To illustrate the use of our results, we show that if the weights of $\Phi$ are standard Gaussian random

variables, then, with high probability, the quantized neural network $\widetilde{\Phi}$ satisfies

$$\frac{\|\Phi(X) - \widetilde{\Phi}(X)\|_F^2}{\mathbb{E}_\Phi \|\Phi(X)\|_F^2} \lesssim \frac{m(\log N_{\max})^{L+1}}{N_{\min}} \tag{2}$$

where we take the expectation $\mathbb{E}_\Phi$ with respect to the weights of $\Phi$, and $N_{\min}$, $N_{\max}$ represent the minimum and maximum layer width of $\Phi$ respectively. We can regard the relative error bound in 2 as a natural generalization of 1.

In Section 4, we consider the finite alphabet case under the random network hypothesis. Denoting by $N_i$ the number of neurons in the $i$-th layer, we show that it suffices to use $b \leq C \log \log \max\{N_{i-1}, N_i\}$ bits to quantize the $i$-th layer while guaranteeing the same error bounds as in the infinite alphabet case.

It is worth noting that we assume that $\Phi$ is equipped with ReLU activation functions, i.e. $\max\{0, x\}$, throughout this paper. This assumption is only made for convenience and concreteness, and we remark that the non-linearities can be replaced by any Lipschitz functions without changing our results, except for the values of constants. For example, suppose the activations are scalar quantized, i.e., rounded to their nearest element in some alphabet. Then, the composition of the activation quantization map and a Lipschitz neural network non-linearity is essentially Lipschitz (up to an additive constant). Moreover, the resulting Lipschitz constant decreases as one uses a finer alphabet. This illustrates that our results easily extend to cover both weight and activation quantization.

Finally, we empirically test the developed method in Appendix H, by quantizing the weights of several neural network architectures that are originally trained for classification tasks on the ImageNet dataset (Deng et al., 2009). The experiments show only a minor loss of accuracy compared to unquantized models.

## 2 STOCHASTIC QUANTIZATION ALGORITHM

In this section, we start with the notation that will be used throughout this paper and then introduce our stochastic quantization algorithm, and show that it can be viewed as a two-stage algorithm. This in turn will simplify its analysis.

### 2.1 NOTATION AND PRELIMINARIES

We denote various positive absolute constants by C, c. We use $a \lesssim b$ as shorthand for $a \leq Cb$, and $a \gtrsim b$ for $a \geq Cb$. For any matrix $A \in \mathbb{R}^{m \times n}$, $\|A\|_{\max}$ denotes $\max_{i,j} |A_{ij}|$.

#### 2.1.1 QUANTIZATION

An $L$-layer perceptron, $\Phi : \mathbb{R}^{N_0} \to \mathbb{R}^{N_L}$, acts on a vector $x \in \mathbb{R}^{N_0}$ via

$$\Phi(x) := \varphi^{(L)} \circ A^{(L)} \circ \cdots \circ \varphi^{(1)} \circ A^{(1)}(x) \tag{3}$$

where each $\varphi^{(i)} : \mathbb{R}^{N_i} \to \mathbb{R}^{N_i}$ is an activation function acting entrywise, and $A^{(i)} : \mathbb{R}^{N_{i-1}} \to \mathbb{R}^{N_i}$ is an affine map given by $A^{(i)}(z) := W^{(i)\top} z + b^{(i)}$. Here, $W^{(i)} \in \mathbb{R}^{N_{i-1} \times N_i}$ is a weight matrix and $b^{(i)} \in \mathbb{R}^{N_i}$ is a bias vector. Since $w^\top x + b = \langle (w, b), (x, 1) \rangle$, the bias term $b^{(i)}$ can simply be treated as an extra row to the weight matrix $W^{(i)}$, so we will henceforth ignore it. For theoretical analysis, we focus on infinite *mid-tread* alphabets, with step-size $\delta$, i.e., alphabets of the form

$$\mathcal{A} = \mathcal{A}_\infty^\delta := \{\pm k\delta : k \in \mathbb{Z}\} \tag{4}$$

and their finite versions, mid-tread alphabets of the form

$$\mathcal{A} = \mathcal{A}_K^\delta := \{\pm k\delta : 0 \leq k \leq K, k \in \mathbb{Z}\}. \tag{5}$$

Given $\mathcal{A} = \mathcal{A}_\infty^\delta$, the associated *stochastic scalar quantizer* $\mathcal{Q}_{\text{StocQ}} : \mathbb{R} \to \mathcal{A}$ randomly rounds every $z \in \mathbb{R}$ to either the minimum or maximum of the interval $[k\delta, (k+1)\delta]$ containing it, in such a way that $\mathbb{E}(\mathcal{Q}_{\text{StocQ}}(z)) = z$. Specifically, we define

$$\mathcal{Q}_{\text{StocQ}}(z) := \begin{cases} \lfloor \frac{z}{\delta} \rfloor \delta & \text{with probability } p \\ \left( \lfloor \frac{z}{\delta} \rfloor + 1 \right)\delta & \text{with probability } 1 - p \end{cases} \tag{6}$$

where $p = 1 - \frac{z}{\delta} + \lfloor \frac{z}{\delta} \rfloor$. If instead of the infinite alphabet, we use $\mathcal{A} = \mathcal{A}_K^\delta$, then whenever $|z| \leq K\delta$, $\mathcal{Q}_{\text{StocQ}}(z)$ is defined via 6 while $\mathcal{Q}_{\text{StocQ}}(z)$ is assigned $-K\delta$ and $K\delta$ if $z < -K\delta$ and $z > K\delta$ respectively. The idea of stochastic quantization and stochastic rounding has a long history (Forsythe, 1959; Barnes et al., 1951) and it has been widely used in signal processing field (Aysal et al., 2008; Wannamaker et al., 2000).

### 2.1.2 ORTHOGONAL PROJECTIONS AND CONVEX ORDERS

Throughout this paper, we will use *orthogonal projections* and the notion of *convex order* (see, e.g., (Shaked & Shanthikumar, 2007)) in our analysis, see Appendix A for their definitions and properties.

## 2.2 SPFQ FUNDAMENTALS

We start with a data set $X \in \mathbb{R}^{m \times N_0}$ with (vectorized) data stored as rows and a pretrained neural network $\Phi$ with weight matrices $W^{(i)} \in \mathbb{R}^{N_{i-1} \times N_i}$ having neurons as their columns. Let $\Phi^{(i)}$, $\widetilde{\Phi}^{(i)}$ denote the original and quantized neural networks up to layer $i$ respectively so that, for example, $\Phi^{(i)}(x) := \varphi^{(i)} \circ W^{(i)} \circ \cdots \circ \varphi^{(1)} \circ W^{(1)}(x)$. Assuming the first $i-1$ layers have been quantized, define the *activations* from $(i-1)$-th layer as

$$X^{(i-1)} := \Phi^{(i-1)}(X) \in \mathbb{R}^{m \times N_{i-1}} \quad \text{and} \quad \widetilde{X}^{(i-1)} := \widetilde{\Phi}^{(i-1)}(X) \in \mathbb{R}^{m \times N_{i-1}}, \tag{7}$$

which also serve as input data for the $i$-th layer. For each neuron $w \in \mathbb{R}^{N_{i-1}}$ in layer $i$, our goal is to construct a quantized vector $q \in \mathcal{A}^{N_{i-1}}$ such that

$$\widetilde{X}^{(i-1)}q = \sum_{t=1}^{N_{i-1}} q_t \widetilde{X}_t^{(i-1)} \approx \sum_{t=1}^{N_{i-1}} w_t X_t^{(i-1)} = X^{(i-1)}w$$

where $X_t^{(i-1)}$, $\widetilde{X}_t^{(i-1)}$ are the $t$-th columns of $X^{(i-1)}$, $\widetilde{X}^{(i-1)}$. Following the GPFQ scheme in (Lybrand & Saab, 2021; Zhang et al., 2023), our algorithm selects $q_t$ sequentially, for $t = 1, 2, \ldots, N_{i-1}$, so that the approximation error of the $t$-th iteration, denoted by

$$u_t := \sum_{j=1}^{t} w_j X_j^{(i-1)} - \sum_{j=1}^{t} q_j \widetilde{X}_j^{(i-1)} \in \mathbb{R}^m, \tag{8}$$

is well-controlled in the $\ell_2$ norm. Specifically, assuming that the first $t-1$ components of $q$ have been determined, the proposed algorithm maintains the error vector $u_{t-1} = \sum_{j=1}^{t-1} (w_j X_j^{(i-1)} - q_j \widetilde{X}_j^{(i-1)})$, and sets $q_t \in \mathcal{A}$ probabilistically depending on $u_{t-1}$, $X_t^{(i-1)}$, and $\widetilde{X}_t^{(i-1)}$. Note that 8 implies

$$u_t = u_{t-1} + w_t X_t^{(i-1)} - q_t \widetilde{X}_t^{(i-1)} \tag{9}$$

and one can get

$$c^* := \arg\min_{c \in \mathbb{R}} \|u_{t-1} + w_t X_t^{(i-1)} - c\widetilde{X}_t^{(i-1)}\|_2^2 = \frac{\langle \widetilde{X}_t^{(i-1)}, u_{t-1} + w_t X_t^{(i-1)} \rangle}{\|\widetilde{X}_t^{(i-1)}\|_2^2}.$$

Hence, a natural design of $q_t \in \mathcal{A}$ is to quantize $c^*$. Instead of using a deterministic quantizer as in (Lybrand & Saab, 2021; Zhang et al., 2023), we apply the stochastic quantizer in 6, that is

$$q_t := \mathcal{Q}_{\text{StocQ}}(c^*) = \mathcal{Q}_{\text{StocQ}}\left( \frac{\langle \widetilde{X}_t^{(i-1)}, u_{t-1} + w_t X_t^{(i-1)} \rangle}{\|\widetilde{X}_t^{(i-1)}\|_2^2} \right). \tag{10}$$

Putting everything together, the stochastic version of GPFQ, namely SPFQ in its basic form, can now be expressed as follows.

$$\begin{cases} u_0 = 0 \in \mathbb{R}^m, \\ q_t = \mathcal{Q}_{\text{StocQ}}\left( \frac{\langle \widetilde{X}_t^{(i-1)}, u_{t-1} + w_t X_t^{(i-1)} \rangle}{\|\widetilde{X}_t^{(i-1)}\|_2^2} \right), \\ u_t = u_{t-1} + w_t X_t^{(i-1)} - q_t \widetilde{X}_t^{(i-1)} \end{cases} \tag{11}$$

where $t$ iterates over $1, 2, \ldots, N_{i-1}$. In particular, the final error vector is

$$u_{N_{i-1}} = \sum_{j=1}^{N_{i-1}} w_j X_j^{(i-1)} - \sum_{j=1}^{N_{i-1}} q_j \widetilde{X}_j^{(i-1)} = X^{(i-1)}w - \widetilde{X}^{(i-1)}q \tag{12}$$

and our goal is to estimate $\|u_{N_{i-1}}\|_2$.

### 2.3 A TWO-PHASE PIPELINE

An essential observation is that 11 can be equivalently decomposed into two phases.

**Phase I**: Given inputs $X^{(i-1)}$, $\widetilde{X}^{(i-1)}$ and neuron $w \in \mathbb{R}^{N_{i-1}}$ for the $i$-th layer, we first align the input data to the layer, by finding a real-valued vector $\widetilde{w} \in \mathbb{R}^{N_{i-1}}$ such that $\widetilde{X}^{(i-1)}\widetilde{w} \approx X^{(i-1)}w$. Similar to our discussion above 10, we adopt the same sequential selection strategy to obtain each $\widetilde{w}_t$ and deduce the following update rules.

$$\begin{cases} \hat{u}_0 = 0 \in \mathbb{R}^m, \\ \widetilde{w}_t = \frac{\langle \widetilde{X}_t^{(i-1)}, \hat{u}_{t-1} + w_t X_t^{(i-1)} \rangle}{\|\widetilde{X}_t^{(i-1)}\|_2^2}, \\ \hat{u}_t = \hat{u}_{t-1} + w_t X_t^{(i-1)} - \widetilde{w}_t \widetilde{X}_t^{(i-1)} \end{cases} \tag{13}$$

where $t = 1, 2 \ldots, N_{i-1}$. Note that the approximation error is given by

$$\hat{u}_{N_{i-1}} = X^{(i-1)}w - \widetilde{X}^{(i-1)}\widetilde{w}. \tag{14}$$

**Phase II**: After getting the new weights $\widetilde{w}$, we quantize $\widetilde{w}$ using SPFQ with input $\widetilde{X}^{(i-1)}$, i.e., finding $\widetilde{q} \in \mathcal{A}^{N_{i-1}}$ such that $\widetilde{X}^{(i-1)}\widetilde{q} \approx \widetilde{X}^{(i-1)}\widetilde{w}$. This process can be summarized as follows. For $t = 1, 2, \ldots, N_{i-1}$,

$$\begin{cases} \widetilde{u}_0 = 0 \in \mathbb{R}^m, \\ \widetilde{q}_t = \mathcal{Q}_{\text{StocQ}}\left(\widetilde{w}_t + \frac{\langle \widetilde{X}_t^{(i-1)}, \widetilde{u}_{t-1} \rangle}{\|\widetilde{X}_t^{(i-1)}\|_2^2}\right), \\ \widetilde{u}_t = \widetilde{u}_{t-1} + (\widetilde{w}_t - \widetilde{q}_t)\widetilde{X}_t^{(i-1)}. \end{cases} \tag{15}$$

Here, the quantization error is

$$\widetilde{u}_{N_{i-1}} = \widetilde{X}^{(i-1)}(\widetilde{w} - \widetilde{q}). \tag{16}$$

**Proposition 2.1.** *Given inputs $X^{(i-1)}$, $\widetilde{X}^{(i-1)}$ and any neuron $w \in \mathbb{R}^{N_{i-1}}$ for the $i$-th layer, the two-phase formulation given by 13 and 15 generate exactly same result as in 11, that is, $\widetilde{q} = q$.*

Proposition 2.1, which is proved in Appendix B, implies that the quantization error 12 for SPFQ can be split into two parts:

$$u_{N_{i-1}} = X^{(i-1)}w - \widetilde{X}^{(i-1)}q = X^{(i-1)}w - \widetilde{X}^{(i-1)}\widetilde{w} + \widetilde{X}^{(i-1)}(\widetilde{w} - q) = \hat{u}_{N_{i-1}} + \widetilde{u}_{N_{i-1}}.$$

Here, the first error term $\hat{u}_{N_{i-1}}$ results from the data alignment in 13 to generate a new "virtual" neuron $\widetilde{w}$ and the second error term $\widetilde{u}_{N_{i-1}}$ is due to the quantization in 15. It follows that

$$\|u_{N_{i-1}}\|_2 = \|\hat{u}_{N_{i-1}} + \widetilde{u}_{N_{i-1}}\|_2 \leq \|\hat{u}_{N_{i-1}}\|_2 + \|\widetilde{u}_{N_{i-1}}\|_2. \tag{17}$$

Thus, we can bound the quantization error for SPFQ by controlling $\|\hat{u}_{N_{i-1}}\|_2$ and $\|\widetilde{u}_{N_{i-1}}\|_2$.

### 2.4 SPFQ VARIANTS

The two-phase formulation of SPFQ provides a flexible framework that allows for the replacement of one or both phases with alternative algorithms. Here, our focus is on replacing the first, "data-alignment", phase to eliminate, or massively reduce, the error bound associated with this step. Indeed, by exploring alternative approaches, one can improve the error bounds of SPFQ, at the expense of increasing the computational complexity. Below, we present two such alternatives to Phase I.

In Section 3 we derive an error bound associated with the second phase of SPFQ, namely quantization, which is independent of the reconstructed neuron $\widetilde{w}$. Thus, to reduce the bound on $\|u_{N_{i-1}}\|_2$

---

**Algorithm 1:** SPFQ

---

**Input:** An $L$-layer neural network $\Phi$ with weight matrices $W^{(i)} \in \mathbb{R}^{N_{i-1} \times N_i}$, input data
$X \in \mathbb{R}^{m \times N_0}$, order $r \in \mathbb{Z}^+$

1 **for** $i = 1$ **to** $L$ **do**

2      Generate $X^{(i-1)} = \Phi^{(i-1)}(X) \in \mathbb{R}^{m \times N_{i-1}}$ and $\widetilde{X}^{(i-1)} = \widetilde{\Phi}^{(i-1)}(X) \in \mathbb{R}^{m \times N_{i-1}}$

3      **if** *perfect data alignment* **then**

4          For each column $w$ of $W^{(i)}$, find a solution $\widetilde{w}$ to 18 (Phase I) and quantize $\widetilde{w}$ via 15 (Phase II)

5      **else if** *approximate data alignment* **then**

6          For each column $w$ of $W^{(i)}$, obtain $\widetilde{w}$ using the $r$-th order data alignment in 13 and 20 (Phase I) then quantize $\widetilde{w}$ via 15 (Phase II)

7      Obtain the quantized $i$-th layer weights $Q^{(i)} \in \mathcal{A}^{N_{i-1} \times N_i}$

**Output:** Quantized neural network $\widetilde{\Phi}$

---

in 17, we can eliminate $\|\hat{u}_{N_{i-1}}\|_2$ by simply choosing $\widetilde{w}$ with $\widetilde{X}^{(i-1)}\widetilde{w} = X^{(i-1)}w$. As this system of equations may admit infinitely many solutions, we opt for one with the minimal $\|\widetilde{w}\|_\infty$. This choice is motivated by the fact that smaller weights can be accommodated by smaller quantization alphabets, resulting in bit savings in practical applications. In other words, we replace Phase I with the optimization problem

$$\min_{\widetilde{w} \in \mathbb{R}^{N_{i-1}}} \quad \|\widetilde{w}\|_\infty$$
$$\text{s.t.} \quad \widetilde{X}^{(i-1)}\widetilde{w} = X^{(i-1)}w. \tag{18}$$

It is not hard to see that 18 can be formulated as a linear program and solved via standard linear programming techniques (Abdelmalek, 1977). Alternatively, powerful tools like Cadzow's method (Cadzow, 1973; 1974) can also be used to solve linearly constrained infinity-norm optimization problems like 18. Cadzow's method has computational complexity $O(m^2 N_{i-1})$, thus is a factor of $m$ more expensive than our original approach but has the advantage of eliminating $\|\hat{u}_{N_{i-1}}\|_2$.

With this modification, one then proceeds with Phase II as before. Given a minimum $\ell_\infty$ solution $\widetilde{w}$ satisfying $\widetilde{X}^{(i-1)}\widetilde{w} = X^{(i-1)}w$, one can quantize it using 15 and obtain $\widetilde{q} \in \mathcal{A}^{N_{i-1}}$. In this case, $\widetilde{q}$ may not be equal to $q$ in 11 and the quantization error becomes

$$X^{(i-1)}w - \widetilde{X}^{(i-1)}\widetilde{q} = \widetilde{X}^{(i-1)}(\widetilde{w} - \widetilde{q}) = \widetilde{u}_{N_{i-1}} \tag{19}$$

where only Phase II is involved. We summarize this version of SPFQ in Algorithm 1.

The second approach we present herein aims to reduce the computational complexity associated with 18. To that end, we generalize the data alignment process in 13 as follows. Let $r \in \mathbb{Z}^+$ and $w \in \mathbb{R}^{N_{i-1}}$. For $t = 1, 2, \ldots, N_{i-1}$, we perform 13 as before. Now however, for $t = N_{i-1} + 1, N_{i-1} + 2, \ldots, rN_{i-1}$, we run

$$\begin{cases} \hat{v}_{t-1} = \hat{u}_{t-1} - w_t X_t^{(i-1)} + \widetilde{w}_t \widetilde{X}_t^{(i-1)}, \\ \widetilde{w}_t = \frac{\langle \widetilde{X}_t^{(i-1)}, \hat{v}_{t-1} + w_t X_t^{(i-1)} \rangle}{\|\widetilde{X}_t^{(i-1)}\|_2^2}, \\ \hat{u}_t = \hat{v}_{t-1} + w_t X_t^{(i-1)} - \widetilde{w}_t \widetilde{X}_t^{(i-1)} \end{cases} \tag{20}$$

Here, we use modulo $N_{i-1}$ indexing for (the subscripts of) $w, \widetilde{w}, X^{(i-1)}$, and $\widetilde{X}^{(i-1)}$. We call the combination of 13 and 20 the $r$-th order data alignment procedure, which costs $O(rmN_{i-1})$ operations. Applying 15 to the output $\widetilde{w}$ as before, the quantization error consists of two parts:

$$X^{(i-1)}w - \widetilde{X}^{(i-1)}\widetilde{q} = X^{(i-1)}w - \widetilde{X}^{(i-1)}\widetilde{w} + \widetilde{X}^{(i-1)}(\widetilde{w} - \widetilde{q}) = \hat{u}_{rN_{i-1}} + \widetilde{u}_{N_{i-1}}. \tag{21}$$

This version of SPFQ with order $r$ is summarized in Algorithm 1. In Section 3, we prove that the data alignment error $\hat{u}_{rN_{i-1}} = X^{(i-1)}w - \widetilde{X}^{(i-1)}\widetilde{w}$ decays exponentially in order $r$.

## 3 ERROR BOUNDS FOR SPFQ WITH INFINITE ALPHABETS

We can now begin analyzing the errors associated with the above variants of SPFQ. The proof of Theorem 3.1 can be found in Appendix D.

**Theorem 3.1.** *Let $\Phi$ be an L-layer neural network as in 3 where the activation function is $\varphi^{(i)}(x) = \rho(x) := \max\{0, x\}$ for $1 \leq i \leq L$. Let $\mathcal{A} = \mathcal{A}_\infty^\delta$ be as in 4 and $p \in \mathbb{N}$.*

*(a) If we quantize $\Phi$ using Algorithm 1 with perfect data alignment, then*

$$\max_{1 \leq j \leq N_L} \|\Phi(X)_j - \widetilde{\Phi}(X)_j\|_2 \leq \sum_{i=0}^{L-1} (2\pi pm\delta^2)^{\frac{L-i}{2}} \left( \prod_{k=i}^{L-1} \log N_k \right)^{\frac{1}{2}} \max_{1 \leq j \leq N_i} \|X_j^{(i)}\|_2 \quad (22)$$

*holds with probability at least $1 - \sum_{i=1}^{L} \frac{\sqrt{2}mN_i}{N_{i-1}^p}$.*

*(b) If we quantize $\Phi$ using Algorithm 1 with approximate data alignment, then*

$$\max_{1 \leq j \leq N_L} \|\Phi(X)_j - \widetilde{\Phi}(X)_j\|_2 \leq$$

$$\sum_{i=0}^{L-1} \delta\sqrt{2\pi pm \log N_i} \max_{1 \leq j \leq N_i} \|X_j^{(i)}\|_2 \prod_{k=i+1}^{L-1} \left( N_k\|W^{(k+1)}\|_{\max}\|P^{(k)}\|_2^{r-1} + \delta\sqrt{2\pi pm \log N_k} \right) \quad (23)$$

*holds with probability at least $1 - \sum_{i=1}^{L} \frac{\sqrt{2}mN_i}{N_{i-1}^p}$. Here, $P^{(k)} = P_{\widetilde{X}_{N_k}^{(k)\perp}} \ldots P_{\widetilde{X}_2^{(k)\perp}} P_{\widetilde{X}_1^{(k)\perp}}$ is defined in Lemma D.2.*

**Remarks on the error bounds.** A few comments are in order regarding the error bounds associated with Theorem 3.1. First, let us consider the difference between the error bounds 22 and 23. As 23 deals with imperfect data alignment, it involves a term that bounds the mismatch between the quantized and unquantized networks. This term is controlled by the quantity $\|P^{(k)}\|_2^{r-1}$, which is expected to be small when the order $r$ is sufficiently large provided $\|P^{(k)}\|_2 < 1$. In other words, one expects this term to be dominated by the error due to quantization. To get a sense for whether this intuition is valid, consider the case where $\widetilde{X}_1^{(k)}, \widetilde{X}_2^{(k)}, \ldots, \widetilde{X}_{N_k}^{(k)}$ are i.i.d. standard Gaussian vectors. Then Lemma C.3 implies that, with high probability,

$$\|P^{(k)}\|_2^{r-1} \lesssim \left( 1 - \frac{c}{m} \right)^{\frac{(r-1)N_k}{10}} = \left( 1 - \frac{c}{m} \right)^{\frac{-m}{c} \cdot \frac{-c(r-1)N_k}{10m}} \leq e^{-\frac{c(r-1)N_k}{10m}}$$

where $c > 0$ is a constant. In this case, $\|P^{(k)}\|_2^{r-1}$ decays exponentially with respect to $r$ with a favorable dependence on the overparametrization $\frac{N}{m}$. In other words, here, even with a small order $r$, the error bounds in 22 and 23 are quite similar.

Keeping this in mind, our next objective is to assess the quality of these error bounds. We will accomplish this by examining the *relative error* connected to the quantization of a neural network. Specifically, we will concentrate on evaluating the relative error associated with 22 since a similar derivation can be applied to 23.

We begin with the observation that both absolute error bounds 22 and 23 in Theorem 3.1 only involve randomness due to the stochastic quantizer $\mathcal{Q}_{\text{StocQ}}$. In particular, there is no randomness assumption on either the weights or the activations. However, to evaluate the relative error, we suppose that each $W^{(i)} \in \mathbb{R}^{N_{i-1} \times N_i}$ has i.i.d. $\mathcal{N}(0, 1)$ entries and $\{W^{(i)}\}_{i=1}^{L}$ are independent. One needs to make an assumption of this type in order to facilitate the calculation, and more importantly, to avoid adversarial scenarios where the weights are chosen to be in the null-space of the data matrix $\widetilde{X}^{(i)}$. We obtain the following corollary which is proved in Appendix E and shows that the relative error decays with the overparametrization of the neural network.

**Corollary 3.2.** *Let $\Phi$ be an L-layer neural network as in 3 where the activation function is $\varphi^{(i)}(x) = \rho(x) := \max\{0, x\}$ for $1 \leq i \leq L$. Suppose the weight matrix $W^{(i)}$ has i.i.d. $\mathcal{N}(0, 1)$ entries and $\{W^{(i)}\}_{i=1}^{L}$ are independent. Let $X \in \mathbb{R}^{m \times N_0}$ be the input data and $X^{(i)} = \Phi^{(i)}(X) \in \mathbb{R}^{m \times N_i}$ be*

the output of the $i$-th layer defined in 7. Then the following inequalities hold.
(a) Let $p \in \mathbb{N}$ with $p \geq 2$. For $1 \leq i \leq L$,

$$\max_{1 \leq j \leq N_i} \|X_j^{(i)}\|_2 \leq (4p)^{\frac{i}{2}} \Big(\prod_{k=1}^{i-1} N_k\Big)^{\frac{1}{2}} \Big(\prod_{k=0}^{i-1} \log N_k\Big)^{\frac{1}{2}} \|X\|_F \tag{24}$$

holds with probability at least $1 - \sum_{k=1}^{i} \frac{2N_k}{N_{k-1}^p}$.
(b) For $1 \leq i \leq L$, we have

$$\mathbb{E}_\Phi \|X^{(i)}\|_F^2 \geq \frac{\|X\|_F^2}{(2\pi)^i} \prod_{k=1}^{i} N_k \tag{25}$$

where $\mathbb{E}_\Phi$ denotes the expectation with respect to the weights of $\Phi$, that is $\{W^{(i)}\}_{i=1}^{L}$.

Under the same conditions of Corollary 3.2 and further assume $N_{\min} \leq N_i \leq N_{\max}$ for all $i$, and $2m \leq N_{\min}$, we have that, with high probability,

$$\frac{\|\Phi(X) - \widetilde{\Phi}(X)\|_F^2}{\mathbb{E}_\Phi \|\Phi(X)\|_F^2} \lesssim \frac{m(\log N_{\max})^{L+1}}{N_{\min}}. \tag{26}$$

This high probability estimate, proved in Appendix E, indicates that the squared error resulting from quantization decays with the overparametrization of the network, relative to the expected squared norm of the neural network's output. It may be possible to replace the expected squared norm by the squared norm itself using another high probability estimate. However, we refrain from doing so as the main objective of this computation was to gain insight into the decay of the relative error in generic settings and the expectation suffices for that purpose.

## 4 ERROR BOUNDS FOR SPFQ WITH FINITE ALPHABETS

Our goal for this section is to relax the assumption that the quantization alphabet used in our algorithms is infinite. We would also like to evaluate the number of elements $2K$ in our alphabet, and thus the number of bits $b := \log_2(K) + 1$ needed for quantizing each layer. Moreover, for simplicity, here we will only consider Algorithm 1 with perfect data alignment. In this setting, to use a finite quantization alphabet, and still obtain theoretical error bounds, we must guarantee that the argument of the stochastic quantizer in 15 remains smaller than the maximal element in the alphabet. Indeed, if that is the case for all $t = 1, ..., N_{i-1}$ then the error bound for our finite alphabet would be identical as for the infinite alphabet. It remains to determine the right size of such a finite alphabet. To that end, we start with Theorem 4.1, which assumes boundedness of all the aligned weights $\widetilde{w}$ in the $i$-th layer, i.e., the solutions of 18, in order to generate an error bound for a finite alphabet of size $K^{(i)} \gtrsim \sqrt{\log \max\{N_{i-1}, N_i\}}$.

**Theorem 4.1.** *Assuming that the first $i - 1$ layers have been quantized, let $X^{(i-1)}, \widetilde{X}^{(i-1)}$ be as in 7. Let $p, K^{(i)} \in \mathbb{N}$ and $\delta > 0$ satisfying $p \geq 3$. Suppose we quantize $W^{(i)}$ using Algorithm 1 with perfect data alignment and $\mathcal{A} = \mathcal{A}_{K^{(i)}}^\delta$, and suppose the resulting aligned weights $\widetilde{W}^{(i)}$ from solving 18 satisfy*

$$\|\widetilde{W}^{(i)}\|_{\max} \leq \frac{1}{2} K^{(i)} \delta. \tag{27}$$

*Then*

$$\max_{1 \leq j \leq N_i} \|X^{(i-1)} W_j^{(i)} - \widetilde{X}^{(i-1)} Q_j^{(i)}\|_2 \leq \delta \sqrt{2\pi pm \log N_{i-1}} \max_{1 \leq j \leq N_{i-1}} \|\widetilde{X}_j^{(i-1)}\|_2 \tag{28}$$

*holds with probability at least $1 - \frac{\sqrt{2}mN_i}{N_{i-1}^p} - \sqrt{2}N_i \sum_{t=2}^{N_{i-1}} \exp\Big(-\frac{(K^{(i)})^2 \|\widetilde{X}_t^{(i-1)}\|_2^2}{8\pi \max_{1 \leq j \leq t-1} \|\widetilde{X}_j^{(i-1)}\|_2^2}\Big).*

Next, in Theorem 4.2, we show that provided the activations $X^{(i-1)}$ and $\widetilde{X}^{(i-1)}$ of the quantized and unquantized networks are sufficiently close, and provided the weights $w$ follow a random distribution, one can guarantee the needed boundedness of the aligned weights $\widetilde{w}$. This allows us to apply Theorem 4.1 and generate an error bound for finite alphabets. Our focus on random weights here

enables us to avoid certain adversarial situations. Indeed, one can construct activations $X^{(i-1)}$ and $\widetilde{X}^{(i-1)}$ that are arbitrarily close to each other, along with adversarial weights $w$ that together lead to $\|\widetilde{w}\|_\infty$ becoming arbitrarily large. We demonstrate this contrived adversarial scenario in Proposition C.9. However, in generic cases represented by random weights, as shown in Theorem 4.2, the bound on $\widetilde{w}$ is not a major issue. Consequently, one can utilize a finite alphabet for quantization as desired. The following results are proved in Appendix G.

**Theorem 4.2.** *Assuming that the first $i-1$ layers have been quantized, let $X^{(i-1)}$, $\widetilde{X}^{(i-1)}$ be as in 7. Suppose the weight matrix $W^{(i)} \in \mathbb{R}^{N_{i-1} \times N_i}$ has i.i.d. $\mathcal{N}(0,1)$ entries and*

$$\|\widetilde{X}^{(i-1)} - X^{(i-1)}\|_2 \leq \epsilon^{(i-1)} \sigma_1^{(i-1)} < \sigma_m^{(i-1)}, \tag{29}$$

*where $\epsilon^{(i-1)} \in (0,1)$, $\sigma_1^{(i-1)}$ and $\sigma_m^{(i-1)}$ are the largest and smallest singular values of $X^{(i-1)}$ respectively. Let $p, K^{(i)} \in \mathbb{N}$ and $\delta > 0$ such that $p \geq 3$ and*

$$K^{(i)}\delta \geq 2\eta^{(i-1)}\sqrt{2p \log N_{i-1}}. \tag{30}$$

*where $\eta^{(i-1)} := \frac{\sigma_1^{(i-1)}}{\sigma_m^{(i-1)} - \epsilon^{(i-1)}\sigma_1^{(i-1)}}$. If we quantize $W^{(i)}$ using Algorithm 1 with perfect data alignment and $\mathcal{A} = \mathcal{A}_{K^{(i)}}^\delta$, then*

$$\max_{1 \leq j \leq N_i} \|X^{(i-1)}W_j^{(i)} - \widetilde{X}^{(i-1)}Q_j^{(i)}\|_2 \leq \delta\sqrt{2\pi pm \log N_{i-1}} \max_{1 \leq j \leq N_{i-1}} \|\widetilde{X}_j^{(i-1)}\|_2 \tag{31}$$

*holds with probability at least $1 - \frac{2N_i}{N_{i-1}^{p-1}} - \frac{\sqrt{2}mN_i}{N_{i-1}^p} - \sqrt{2}N_i \sum_{t=2}^{N_{i-1}} \exp\left(-\frac{(K^{(i)})^2\|\widetilde{X}_t^{(i-1)}\|_2^2}{8\pi \max_{1 \leq j \leq t-1}\|\widetilde{X}_j^{(i-1)}\|_2^2}\right)$.*

Now we are about to approximate the number of bits needed for guaranteeing the derived bounds. Note that, in Theorem 4.2, we achieved the same error bound 31 as in Lemma D.1, choosing proper $\epsilon^{(i-1)} \in (0,1)$ and $K^{(i)} \in \mathbb{N}$ such that 29 and 30 are satisfied and the associated probability in 31 is positive. This implies that the error bounds we obtained in Section 3 remain valid for our finite alphabets as well. In particular, by a similar argument we used to obtain 79, one can get the following approximations

$$\frac{\|\widetilde{X}^{(i-1)} - X^{(i-1)}\|_F^2}{\|X^{(i-1)}\|_F^2} \lesssim \left(\prod_{k=0}^{i-1} \log N_k\right) \sum_{j=0}^{i-2} \prod_{k=j}^{i-2} \frac{m}{N_k}.$$

Due to $\|X^{(i-1)}\|_F \leq \sqrt{m}\|X^{(i-1)}\|_2$ and $\|\widetilde{X}^{(i-1)} - X^{(i-1)}\|_2 \leq \|\widetilde{X}^{(i-1)} - X^{(i-1)}\|_F$, we have

$$\frac{\|\widetilde{X}^{(i-1)} - X^{(i-1)}\|_2^2}{\|X^{(i-1)}\|_2^2} \leq \frac{m\|\widetilde{X}^{(i-1)} - X^{(i-1)}\|_F^2}{\|X^{(i-1)}\|_F^2} \lesssim m\left(\prod_{k=0}^{i-1} \log N_k\right) \sum_{j=0}^{i-2} \prod_{k=j}^{i-2} \frac{m}{N_k}.$$

If $\prod_{k=j}^{i-2} N_k \gtrsim m^{i-j} \prod_{k=0}^{i-1} \log N_k$ for $0 \leq j \leq i-2$, then it is possible to choose $\epsilon^{(i-1)} \in (0,1)$ such that 29 holds. Moreover, since $\sigma_m^{(i-1)} \leq \sigma_1^{(i-1)}$, we have $\eta^{(i-1)} = \frac{\sigma_1^{(i-1)}}{\sigma_m^{(i-1)} - \epsilon^{(i-1)}\sigma_1^{(i-1)}} \geq (1 - \epsilon^{(i-1)})^{-1}$ and thus 30 becomes

$$K^{(i)} \geq 2\delta^{-1}(1 - \epsilon^{(i-1)})^{-1}\sqrt{2p \log N_{i-1}} \gtrsim \sqrt{\log N_{i-1}}. \tag{32}$$

Assuming columns of $\widetilde{X}^{(i-1)}$ are similar in the sense of

$$\max_{1 \leq j \leq t-1} \|\widetilde{X}_j^{(i-1)}\|_2 \lesssim \sqrt{\log N_{i-1}}\|\widetilde{X}_t^{(i-1)}\|_2, \quad 2 \leq t \leq N_{i-1},$$

we obtain that 31 holds with probability exceeding

$$1 - \frac{2N_i}{N_{i-1}^{p-1}} - \frac{\sqrt{2}mN_i}{N_{i-1}^p} - \sqrt{2}N_i \sum_{t=2}^{N_{i-1}} \exp\left(-\frac{(K^{(i)})^2\|\widetilde{X}_t^{(i-1)}\|_2^2}{8\pi \max_{1 \leq j \leq t-1}\|\widetilde{X}_j^{(i-1)}\|_2^2}\right)$$

$$\geq 1 - \frac{2N_i}{N_{i-1}^{p-1}} - \frac{\sqrt{2}mN_i}{N_{i-1}^p} - \sqrt{2}N_{i-1}N_i \exp\left(-\frac{(K^{(i)})^2}{8\pi \log N_{i-1}}\right). \tag{33}$$

To make 33 positive, we have

$$K^{(i)} \gtrsim \log \max\{N_{i-1}, N_i\}. \tag{34}$$

It follows from 32 and 33 that, in the $i$th layer, we only need a number of bits $b^{(i)}$ that satisfies

$$b^{(i)} \geq \log_2 K^{(i)} + 1 \gtrsim \log_2 \log \max\{N_{i-1}, N_i\}$$

to guarantee the performance of our quantization method using finite alphabets.

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

## A  PROPERTIES OF ORTHOGONAL PROJECTIONS AND CONVEX ORDERS

### A.1  ORTHOGONAL PROJECTIONS

Given a subspace $S \subseteq \mathbb{R}^m$, we denote by $S^{\perp}$ its orthogonal complement in $\mathbb{R}^m$, and by $P_S$ the orthogonal projection of $\mathbb{R}^m$ onto $S$. In particular, if $z \in \mathbb{R}^m$ is a nonzero vector, then we use $P_z$ and $P_{z^{\perp}}$ to represent orthogonal projections onto $\mathrm{span}(z)$ and $\mathrm{span}(z)^{\perp}$ respectively. Hence, for any $x \in \mathbb{R}^m$, we have

$$P_z(x) = \frac{\langle z, x \rangle z}{\|z\|_2^2}, \quad x = P_z(x) + P_{z^{\perp}}(x), \quad \text{and} \quad \|x\|_2^2 = \|P_z(x)\|_2^2 + \|P_{z^{\perp}}(x)\|_2^2. \tag{35}$$

Throughout this paper, we will also use $P_z$ and $P_{z^{\perp}}$ to denote the associated matrix representations satisfying

$$P_z x = \frac{zz^{\top}}{\|z\|_2^2} x \quad \text{and} \quad P_{z^{\perp}} x = \left( I - \frac{zz^{\top}}{\|z\|_2^2} \right) x. \tag{36}$$

### A.2  CONVEX ORDER

We now introduce the concept of *convex order*, which is essential for our theoretical analysis. Throughout this section, $\overset{d}{=}$ denotes equality in distribution.

**Definition A.1.** Let $X, Y$ be $n$-dimensional random vectors such that

$$\mathbb{E} f(X) \leq \mathbb{E} f(Y) \tag{37}$$

holds for all convex functions $f : \mathbb{R}^n \to \mathbb{R}$, provided the expectations exist. Then $X$ is said to be smaller than $Y$ in the *convex order*, denoted by $X \leq_{\mathrm{cx}} Y$.

For $i = 1, 2, \ldots, n$, define functions $\phi_i(x) := x_i$ and $\psi_i(x) := -x_i$. Since both $\phi_i(x)$ and $\psi_i(x)$ are convex, substituting them into 37 yields $\mathbb{E} X_i = \mathbb{E} Y_i$ for all $i$. Therefore, we obtain

$$X \leq_{\mathrm{cx}} Y \implies \mathbb{E} X = \mathbb{E} Y. \tag{38}$$

Clearly, according to Definition A.1, $X \leq_{\mathrm{cx}} Y$ only depends on the respective distributions of $X$ and $Y$. It can be easily seen that the relation $\leq_{\mathrm{cx}}$ satisfies reflexivity and transitivity. In other words, one has $X \leq_{\mathrm{cx}} X$ and that if $X \leq_{\mathrm{cx}} Y$ and $Y \leq_{\mathrm{cx}} Z$, then $X \leq_{\mathrm{cx}} Z$. The convex order defined in Definition A.1 is also called *mean-preserving spread* (Rothschild & Stiglitz, 1970; Machina & Pratt, 1997), which is a special case of *second-order stochastic dominance* (Hadar & Russell, 1969; Hanoch & Levy, 1975; Shaked & Shanthikumar, 2007). A well-known result is that the convex order can be characterized by a *coupling* of $X$ and $Y$, i.e. constructing $X$ and $Y$ on the same probability space.

**Theorem A.2** (Theorem 7.A.1 in (Shaked & Shanthikumar, 2007)). *The random vectors $X$ and $Y$ satisfy $X \leq_{\mathrm{cx}} Y$ if and only if there exist two random vectors $\hat{X}$ and $\hat{Y}$, defined on the same probability space, such that $\hat{X} \stackrel{d}{=} X$, $\hat{Y} \stackrel{d}{=} Y$, and $\mathbb{E}(\hat{Y}|\hat{X}) = \hat{X}$.*

In Theorem A.2, $\mathbb{E}(\hat{Y}|\hat{X}) = \hat{X}$ implies $\mathbb{E}(\hat{Y} - \hat{X}|\hat{X}) = 0$. Let $\hat{Z} := \hat{Y} - \hat{X}$. Then we have $\hat{Y} = \hat{X} + \hat{Z}$ with $\mathbb{E}(\hat{Z}|\hat{X}) = 0$. Thus, one can obtain $\hat{Y}$ by first sampling $\hat{X}$, and then adding a mean 0 random vector $\hat{Z}$ whose distribution may depend on the sampled $\hat{X}$. Based on this important observation, the following result gives necessary and sufficient conditions for the comparison of multivariate normal random vectors, see e.g. Example 7.A.13 in (Shaked & Shanthikumar, 2007).

**Lemma A.3.** *Consider multivariate normal distributions $\mathcal{N}(\mu_1, \Sigma_1)$ and $\mathcal{N}(\mu_2, \Sigma_2)$. Then*

$$\mathcal{N}(\mu_1, \Sigma_1) \leq_{\mathrm{cx}} \mathcal{N}(\mu_2, \Sigma_2) \iff \mu_1 = \mu_2 \quad \text{and} \quad \Sigma_1 \preceq \Sigma_2.$$

*Proof.* ($\Rightarrow$) Suppose that $X \sim \mathcal{N}(\mu_1, \Sigma_1)$ and $Y \sim \mathcal{N}(\mu_2, \Sigma_2)$ such that $X \leq_{\mathrm{cx}} Y$. By 38, we have $\mu_1 = \mu_2$. Let $a \in \mathbb{R}^n$ and define $f(x) := (a^\top x - a^\top \mu_1)^2$. Since $f(x)$ is convex, one can get

$$a^\top \Sigma_1 a = \mathrm{Var}(a^\top X) = \mathbb{E}f(X) \leq \mathbb{E}f(Y) = \mathrm{Var}(a^\top Y) = a^\top \Sigma_2 a.$$

Since this inequality holds for arbitrary $a \in \mathbb{R}^n$, we obtain $\Sigma_1 \preceq \Sigma_2$.

($\Leftarrow$) Conversely, assume that $\mu_1 = \mu_2$ and $\Sigma_1 \preceq \Sigma_2$. Let $X \sim \mathcal{N}(\mu_1, \Sigma_1)$ and $Z \sim \mathcal{N}(0, \Sigma_2 - \Sigma_1)$ be independent. Construct a random vector $Y := X + Z$. Then $Y \sim \mathcal{N}(\mu_2, \Sigma_2)$ and $\mathbb{E}(Y|X) = \mathbb{E}(X + Z|X) = X + \mathbb{E}Z = X$. Following Theorem A.2, $\mathcal{N}(\mu_1, \Sigma_1) \leq_{\mathrm{cx}} \mathcal{N}(\mu_2, \Sigma_2)$ holds. $\square$

Moreover, the convex order is preserved under affine transformations.

**Lemma A.4.** *Suppose that $X$, $Y$ are $n$-dimensional random vectors satisfying $X \leq_{\mathrm{cx}} Y$. Let $A \in \mathbb{R}^{m \times n}$ and $b \in \mathbb{R}^m$. Then $AX + b \leq_{\mathrm{cx}} AY + b$.*

*Proof.* Let $f : \mathbb{R}^n \to \mathbb{R}$ be any convex function. Since $g(x) := f(Ax + b)$ is a composition of convex function $f(x)$ and a linear map, $g(x)$ is also convex. As $X \leq_{\mathrm{cx}} Y$, we now have

$$\mathbb{E}f(AX + b) = \mathbb{E}g(X) \leq \mathbb{E}g(Y) = \mathbb{E}f(AY + b),$$

so $AX + b \leq_{\mathrm{cx}} AY + b$. $\square$

The following results, which will also be useful to us, were proved in Section 2 of (Alweiss et al., 2021).

**Lemma A.5.** *Consider random vectors $X$, $Y$, $W$, and $Z$. Let $X$ and $Y$ live on the same probability space, and let $W$ and $Z$ be independent. Suppose that $X \leq_{\mathrm{cx}} W$ and $(Y - X)|X \leq_{\mathrm{cx}} Z$. Then $Y \leq_{\mathrm{cx}} W + Z$.*

**Lemma A.6.** *Let $X$ be a real-valued random variable with $\mathbb{E}X = 0$ and $|X| \leq C$. Then $X \leq_{\mathrm{cx}} \mathcal{N}\left(0, \frac{\pi C^2}{2}\right)$.*

Applying Lemma A.5 inductively, one can show that the convex order is closed under convolutions.

**Lemma A.7.** *Let $X_1, X_2, \ldots, X_m$ be a set of independent random vectors and let $Y_1, Y_2, \ldots, Y_m$ be another set of independent random vectors. If $X_i \leq_{\mathrm{cx}} Y_i$ for $1 \leq i \leq m$, then*

$$\sum_{i=1}^{m} X_i \leq_{\mathrm{cx}} \sum_{i=1}^{m} Y_i. \tag{39}$$

*Proof.* We will prove 39 by induction on $m$. The case $m = 1$ is trivial. Assume that the lemma holds for $m - 1$ with $m \geq 2$, and let us prove it for $m$. Applying Lemma A.5 for $X = X_m$, $Y = \sum_{i=1}^{m} X_i$, $W = Y_m$, and $Z = \sum_{i=1}^{m-1} Y_i$, inequality 39 follows. $\square$

# B  PROOF OF PROPOSITION 2.1

*Proof.* We proceed by induction on the iteration index $t$. If $t = 1$, then 13, 15 and 11 imply that

$$\widetilde{q}_1 = \mathcal{Q}_{\text{StocQ}}(\widetilde{w}_1) = \mathcal{Q}_{\text{StocQ}}\Big(\frac{\langle \widetilde{X}_1^{(i-1)}, w_1 X_1^{(i-1)}\rangle}{\|\widetilde{X}_1^{(i-1)}\|_2^2}\Big) = q_1.$$

For $t \geq 2$, assume $\widetilde{q}_j = q_j$ for $1 \leq j \leq t-1$ and we aim to prove $\widetilde{q}_t = q_t$. Note that $\hat{u}_{t-1} = \sum_{j=1}^{t-1}(w_j X_j - \widetilde{w}_j \widetilde{X}_j)$ and $\widetilde{u}_{t-1} = \sum_{j=1}^{t-1}(\widetilde{w}_j \widetilde{X}_j - \widetilde{q}_j \widetilde{X}_j) = \sum_{j=1}^{t-1}(\widetilde{w}_j \widetilde{X}_j - q_j \widetilde{X}_j)$ by our induction hypothesis. It follows that $\hat{u}_{t-1} + \widetilde{u}_{t-1} = \sum_{j=1}^{t-1}(w_j X_j - q_j \widetilde{X}_j) = u_{t-1}$. Thus, we get

$$\widetilde{q}_t = \mathcal{Q}_{\text{StocQ}}\Big(\frac{\langle \widetilde{X}_t^{(i-1)}, \widetilde{u}_{t-1} + \hat{u}_{t-1} + w_t X_t^{(i-1)}\rangle}{\|\widetilde{X}_t^{(i-1)}\|_2^2}\Big) = \mathcal{Q}_{\text{StocQ}}\Big(\frac{\langle \widetilde{X}_t^{(i-1)}, u_{t-1} + w_t X_t^{(i-1)}\rangle}{\|\widetilde{X}_t^{(i-1)}\|_2^2}\Big) = q_t.$$

This establishes $\widetilde{q} = q$ and completes the proof. □

# C  USEFUL LEMMATA

## C.1  CONCENTRATION INEQUALITIES

The following two lemmata are essential for the approximation of quantization error bounds. The proof techniques follow (Alweiss et al., 2021).

**Lemma C.1.** *Let $\alpha > 0$ and $z_1, z_2, \ldots, z_d \in \mathbb{R}^m$ be nonzero vectors. Let $M_0 = 0$. For $1 \leq t \leq d$, define $M_t \in \mathbb{R}^{m \times m}$ inductively as*

$$M_t := P_{z_t^\perp} M_{t-1} P_{z_t^\perp} + \alpha z_t z_t^\top$$

*where $P_{z_t^\perp} = I - \frac{z_t z_t^\top}{\|z_t\|_2^2}$ is the orthogonal projection as in 36. Then*

$$M_t \preceq \beta_t I \tag{40}$$

*holds for all $t$, where $\beta_t := \alpha \max_{1 \leq j \leq t} \|z_j\|_2^2$.*

*Proof.* We proceed by induction on $t$. If $t = 1$, then $M_1 = \alpha z_1 z_1^\top$. By Cauchy-Schwarz inequality, for any $x \in \mathbb{R}^m$, we get

$$x^\top M_1 x = \alpha \langle z_1, x\rangle^2 \leq \alpha \|z_1\|_2^2 \|x\|_2^2 = \beta_1 \|x\|_2^2 = x^\top(\beta_1 I)x.$$

It follows that $M_1 \preceq \beta_1 I$. Now, assume that 40 holds for $t-1$ with $t \geq 2$. Then we have

$$\begin{aligned}
M_t &= P_{z_t^\perp} M_{t-1} P_{z_t^\perp} + \alpha z_t z_t^\top \\
&\preceq \beta_{t-1} P_{z_t^\perp}^2 + \alpha z_t z_t^\top && \text{(by assumption } M_{t-1} \preceq \beta_{t-1} I) \\
&\preceq \beta_t P_{z_t^\perp} + \alpha z_t z_t^\top && \text{(since } P_{z_t^\perp}^2 = P_{z_t^\perp} \text{ and } \beta_{t-1} \leq \beta_t) \\
&= \beta_t I + (\alpha \|z_t\|_2^2 - \beta_t)\frac{z_t z_t^\top}{\|z_t\|_2^2} && \text{(using 36)} \\
&\preceq \beta_t I && (\text{as } \beta_t = \alpha \max_{1 \leq j \leq t} \|z_j\|_2^2).
\end{aligned}$$

This completes the proof. □

**Lemma C.2.** *Let $X$ be an $n$-dimensional random vector such that $X \leq_{\text{cx}} \mathcal{N}(\mu, \sigma^2 I)$, and let $\alpha > 0$. Then*

$$\mathrm{P}\Big(\|X - \mu\|_\infty \leq \alpha\Big) \geq 1 - \sqrt{2}n e^{-\frac{\alpha^2}{4\sigma^2}}.$$

*In particular, if $\alpha = 2\sigma\sqrt{\log(\sqrt{2}n/\gamma)}$ with $\gamma \in (0, 1]$, we have*

$$P\Big(\|X - \mu\|_\infty \leq 2\sigma\sqrt{\log(\sqrt{2}n/\gamma)}\Big) \geq 1 - \gamma.$$

*Proof.* Let $x \in \mathbb{R}^n$ with $\|x\|_2 \leq 1$. Since $X \leq_{\mathrm{cx}} \mathcal{N}(\mu, \sigma^2 I)$, by Lemma A.3 and Lemma A.4, we get

$$\frac{\langle X - \mu, x \rangle}{\sigma} \leq_{\mathrm{cx}} \mathcal{N}(0, \|x\|_2^2) \leq_{\mathrm{cx}} \mathcal{N}(0, 1).$$

Then we have

$$\mathbb{E} e^{\frac{\langle X - \mu, x \rangle^2}{4\sigma^2}} \leq \mathbb{E}_{Z \sim \mathcal{N}(0,1)} e^{Z^2/4} = \sqrt{2}.$$

where we used Definition A.1 on the convex function $f(x) = e^{x^2/4}$. By Markov's inequality and the inequality above, we conclude that

$$\begin{aligned}
\mathrm{P}(|\langle X - \mu, x \rangle| \geq \alpha) &= \mathrm{P}\Big( e^{\frac{\langle X - \mu, x \rangle^2}{4\sigma^2}} \geq e^{\frac{\alpha^2}{4\sigma^2}} \Big) \\
&\leq e^{-\frac{\alpha^2}{4\sigma^2}} \mathbb{E} e^{\frac{\langle X - \mu, x \rangle^2}{4\sigma^2}} \\
&\leq \sqrt{2} e^{-\frac{\alpha^2}{4\sigma^2}}.
\end{aligned}$$

Finally, by a union bound over the standard basis vectors $x = e_1, e_2, \ldots, e_n$, we have

$$\mathrm{P}\Big( \|X - \mu\|_\infty \leq \alpha \Big) \geq 1 - \sqrt{2} n e^{-\frac{\alpha^2}{4\sigma^2}}.$$

$\square$

Lemma C.3 below will be used to illustrate the behavior of the norm of the successive projection operator appearing in Theorem 3.1 in the case of random inputs.

**Lemma C.3.** *Let $X_1, X_2, \ldots, X_N$ be i.i.d. random vectors drawn from $\mathcal{N}(0, I_m)$. Let $N \geq 10$ and $P := P_{X_N^\perp} \ldots P_{X_2^\perp} P_{X_1^\perp} \in \mathbb{R}^{m \times m}$. Then*

$$\mathrm{P}\Big( \|P\|_2^2 \leq 4 \big( 1 - \frac{c}{m} \big)^{\lfloor \frac{N}{5} \rfloor} \Big) \geq 1 - 5^m e^{-\frac{N}{5}} \tag{41}$$

*where $c > 0$ is an absolute constant.*

*Proof.* This proof is based on an $\epsilon$-net argument. By the definition of $\|P\|_2$, we need to bound $\|Pz\|_2$ for all vectors $z \in \mathbb{S}^{m-1}$. To this end, we will cover the unit sphere using small balls with radius $\epsilon$, establish tight control of $\|Pz\|_2$ for every fixed vector $z$ from the net, and finally take a union bound over all vectors in the net.

We first set up an $\epsilon$-net. Choosing $\epsilon = \frac{1}{2}$, according to Corollary 4.2.13 in (Vershynin, 2018), we can find an $\epsilon$-net $\mathcal{D} \subseteq \mathbb{S}^{m-1}$ such that

$$\mathbb{S}^{m-1} \subseteq \bigcup_{z \in \mathcal{D}} B(z, \epsilon) \quad \text{and} \quad |\mathcal{D}| \leq \Big( 1 + \frac{2}{\epsilon} \Big)^m = 5^m. \tag{42}$$

Here, $B(z, \epsilon)$ represents the closed ball centered at $z$ and with radius $\epsilon$, and $|\mathcal{D}|$ is the cardinality of $\mathcal{D}$. Moreover, we have (see Lemma 4.4.1 in (Vershynin, 2018))

$$\|P\|_2 \leq \frac{1}{1 - \epsilon} \max_{z \in \mathcal{D}} \|Pz\|_2 = 2 \max_{z \in \mathcal{D}} \|Pz\|_2. \tag{43}$$

Next, let $\beta \geq 1$, $\gamma > 0$, and $z \in \mathbb{S}^{m-1}$. Applying 35 and setting $\xi \sim \mathcal{N}(0, I_m)$, for $1 \leq j \leq N$, we obtain

$$\begin{aligned}
\mathrm{P}\Big( \|P_{X_j^\perp}(z)\|_2^2 \geq 1 - \gamma \Big) &= \mathrm{P}\Big( \|P_{X_j}(z)\|_2^2 \leq \gamma \Big) \\
&= \mathrm{P}\Big( \Big\langle \frac{X_j}{\|X_j\|_2}, z \Big\rangle^2 \leq \gamma \Big) \\
&= \mathrm{P}\Big( \Big\langle \frac{\xi}{\|\xi\|_2}, z \Big\rangle^2 \leq \gamma \Big).
\end{aligned}$$

By rotation invariance of the normal distribution, we may assume without loss of generality that $z = e_1 := (1, 0, \ldots, 0) \in \mathbb{R}^m$. It follows that

$$
\begin{aligned}
\mathrm{P}\Big(\|P_{X_j^\perp}(z)\|_2^2 \geq 1 - \gamma\Big) &= \mathrm{P}\Big(\frac{\xi_1^2}{\|\xi\|_2^2} \leq \gamma\Big) \\
&= \mathrm{P}\Big(\frac{\xi_1^2}{\|\xi\|_2^2} \leq \gamma,\ \|\xi\|_2^2 \leq \beta m\Big) + \mathrm{P}\Big(\frac{\xi_1^2}{\|\xi\|_2^2} \leq \gamma,\ \|\xi\|_2^2 > \beta m\Big) \\
&\leq \mathrm{P}(\xi_1^2 \leq \beta \gamma m) + \mathrm{P}(\|\xi\|_2^2 \geq \beta m) \\
&\leq \sqrt{\frac{2\beta\gamma m}{\pi}} + 2\exp(-c'm(\sqrt{\beta} - 1)^2).
\end{aligned}
\tag{44}
$$

In the last step, we controlled the probability via

$$
\mathrm{P}(\xi_1^2 \leq \beta\gamma m) = \int_{-\sqrt{\beta\gamma m}}^{\sqrt{\beta\gamma m}} \frac{1}{\sqrt{2\pi}} e^{-\frac{1}{2}x^2}\, dx \leq \frac{1}{\sqrt{2\pi}} \int_{-\sqrt{\beta\gamma m}}^{\sqrt{\beta\gamma m}} 1\, dx = \sqrt{\frac{2\beta\gamma m}{\pi}},
$$

and used the concentration of the norm (see Theorem 3.1.1 in (Vershynin, 2018)):

$$
\mathrm{P}(\|\xi\|_2^2 \geq \beta m) \leq 2\exp(-c'm(\sqrt{\beta} - 1)^2), \quad \beta \geq 1,
$$

where $c' > 0$ is an absolute constant. In 44, picking $\beta = (\sqrt{\frac{3}{c'}} + 1)^2$ and $\gamma = \frac{1}{12\beta m} = \frac{c}{m}$ with $c := \frac{1}{12}(\sqrt{\frac{3}{c'}} + 1)^{-2}$, we have that

$$
\tau := \mathrm{P}\Big(\|P_{X_j^\perp}(z)\|_2^2 \leq 1 - \frac{c}{m}\Big) \geq 1 - \sqrt{\frac{1}{6\pi}} - 2e^{-3m} \geq 1 - \sqrt{\frac{1}{6\pi}} - 2e^{-3} \geq \frac{2}{3}
\tag{45}
$$

holds for all $1 \leq j \leq N$ and $z \in \mathbb{S}^{m-1}$. So each orthogonal projection $P_{X_j^\perp}$ can reduce the squared norm of a vector to at most $1 - \frac{c}{m}$ ratio with probability $\tau$. Fix $z \in \mathcal{D}$. Since $X_1, X_2, \ldots, X_n$ are independent, we have

$$
\begin{aligned}
\mathrm{P}\Big(\|Pz\|_2^2 \geq \Big(1 - \frac{c}{m}\Big)^{\lfloor \frac{N}{5} \rfloor}\Big) &\leq \sum_{k=0}^{\lfloor \frac{N}{5} \rfloor} \binom{N}{k} \tau^k (1-\tau)^{N-k} \\
&\leq \sum_{k=0}^{\lfloor \frac{N}{5} \rfloor} \binom{N}{k} (1-\tau)^{N-k} && \text{(since } \tau \leq 1\text{)} \\
&\leq (1-\tau)^{N - \lfloor \frac{N}{5} \rfloor} \sum_{k=0}^{\lfloor \frac{N}{5} \rfloor} \binom{N}{k} \\
&\leq \Big(\frac{1}{3}\Big)^{N - \lfloor \frac{N}{5} \rfloor} \sum_{k=0}^{\lfloor \frac{N}{5} \rfloor} \binom{N}{k} && \text{(by 45)} \\
&\leq \Big(\frac{1}{3}\Big)^{N - \lfloor \frac{N}{5} \rfloor} \Big(\frac{eN}{\lfloor \frac{N}{5} \rfloor}\Big)^{\lfloor \frac{N}{5} \rfloor} && \text{(due to } \sum_{k=0}^{l} \binom{n}{k} \leq \Big(\frac{en}{l}\Big)^l\text{).} \tag{46}
\end{aligned}
$$

Since $\frac{N}{5} - 1 < \lfloor \frac{N}{5} \rfloor \leq \frac{N}{5}$ and $N \geq 10$, we have

$$
\Big(\frac{1}{3}\Big)^{N - \lfloor \frac{N}{5} \rfloor} \Big(\frac{eN}{\lfloor \frac{N}{5} \rfloor}\Big)^{\lfloor \frac{N}{5} \rfloor} \leq \Big(\frac{1}{3}\Big)^{\frac{4N}{5}} \Big(\frac{eN}{\frac{N}{5} - 1}\Big)^{\frac{N}{5}} = \Big(\frac{1}{81} \cdot \frac{5e}{1 - \frac{5}{N}}\Big)^{\frac{N}{5}} \leq \Big(\frac{10e}{81}\Big)^{\frac{N}{5}} \leq e^{-\frac{N}{5}}.
$$

Plugging this into 46, we deduce that

$$
\mathrm{P}\Big(\|Pz\|_2^2 \leq \Big(1 - \frac{c}{m}\Big)^{\lfloor \frac{N}{5} \rfloor}\Big) \geq 1 - e^{-\frac{N}{5}}.
$$

holds for all $z \in \mathcal{D}$. By a union bound over $|\mathcal{D}| \leq 5^m$ points, we obtain

$$
\mathrm{P}\Big(\max_{z \in \mathcal{D}} \|Pz\|_2^2 \leq \Big(1 - \frac{c}{m}\Big)^{\lfloor \frac{N}{5} \rfloor}\Big) \geq 1 - 5^m e^{-\frac{N}{5}}.
\tag{47}
$$

Then 41 follows immediately from 43 and 47. □

Moreover, we present the following result on concentration of (Gaussian) measure inequality for Lipschitz functions, which will be used in the proofs to control the effect of the non-linear function $\rho$ that appears in the neural network.

**Lemma C.4.** *Consider an $n$-dimensional random vector $X \sim \mathcal{N}(0, I)$ and a Lipschitz function $f : \mathbb{R}^n \to \mathbb{R}$ with Lipschitz constant $L_f > 0$, that is $|f(x) - f(y)| \leq L_f \|x - y\|_2$ for all $x, y \in \mathbb{R}^n$. Then, for all $\alpha \geq 0$,*

$$\mathrm{P}(|f(X) - \mathbb{E}f(X)| \geq \alpha) \leq 2\exp\Big(-\frac{\alpha^2}{2L_f^2}\Big).$$

A proof of Lemma C.4 can be found in Chapter 8 of (Foucart & Rauhut, 2013). Further, the following result provides a lower bound for the expected activation associated with inputs drawn from a Gaussian distribution. This will be used to illustrate the relative error associated with SPFQ.

**Proposition C.5.** *Let $\rho(x) := \max\{0, x\}$ be the ReLU activation function, acting elementwise, and let $X \sim \mathcal{N}(0, \Sigma)$. Then*

$$\mathbb{E}\|\rho(X)\|_2 \geq \sqrt{\frac{\mathrm{tr}(\Sigma)}{2\pi}}.$$

To start the proof of Proposition C.5, we need the following two lemmas. While these results are likely to be known, we could not find proofs in the literature so we include the argument for completeness.

**Lemma C.6.** *Let $\mathcal{S}$ denote the convex set of all positive semidefinite matrices $A$ in $\mathbb{R}^{n \times n}$ with $\mathrm{tr}(A) = 1$. Then the extreme points of $\mathcal{S}$ are exactly the rank-1 matrices of the form $uu^\top$ where $u$ is a unit vector in $\mathbb{R}^n$.*

*Proof.* We first let $A \in \mathcal{S}$ be an extreme point of $\mathcal{S}$ and assume $\mathrm{rank}(A) = r > 1$. Since $A$ is positive semidefinite, the spectral decomposition of $A$ yields $A = \sum_{i=1}^{r} \lambda_i u_i u_i^\top$ where $\lambda_i > 0$ and $\|u_i\|_2 = 1$ for $1 \leq i \leq r$. Then $A$ can be rewritten as

$$A = \Big(\sum_{j=1}^{r-1} \lambda_j\Big) B + \lambda_r u_r u_r^\top$$

where $B = \sum_{i=1}^{r-1} \frac{\lambda_i}{\sum_{j=1}^{r-1} \lambda_j} u_i u_i^\top$. Note that $B$ and $u_r u_r^\top$ are distinct positive semidefinite matrices with $\mathrm{tr}(B) = \mathrm{tr}(u_r u_r^\top) = 1$, and $\sum_{j=1}^{r} \lambda_j = \mathrm{tr}(A) = 1$. Thus, $B, u_r u_r^\top \in \mathcal{S}$ and $A$ is in the open line segment joining $B$ and $u_r u_r^\top$, which is a contradiction. So any extreme point of $\mathcal{S}$ is a rank-1 matrix of the form $A = uu^\top$ with $\|u\|_2 = 1$.

Conversely, consider any rank-1 matrix $A = uu^\top$ with $\|u\|_2 = 1$. Then we have $A \in \mathcal{S}$. Assume that $A$ lies in an open segment in $\mathcal{S}$ connecting two distinct matrices $A_1, A_2 \in \mathcal{S}$, that is

$$A = \alpha_1 A_1 + \alpha_2 A_2 \tag{48}$$

where $\alpha_1 + \alpha_2 = 1$ and $0 < \alpha_1 \leq \alpha_2$. Additionally, for any $x \in \ker(A)$, we have

$$0 = x^\top A x = \alpha_1 x^\top A_1 x + \alpha_2 x^\top A_2 x \tag{49}$$

and thus $A_1 x = A_2 x = 0$. It implies $\ker(A) \subseteq \ker(A_1) \cap \ker(A_2)$. By the rank–nullity theorem, we get $1 = \mathrm{rank}(A) \geq \max\{\mathrm{rank}(A_1), \mathrm{rank}(A_2)\}$. Since $A_1$ and $A_2$ are distinct matrices in $\mathcal{S}$, we have $\mathrm{rank}(A_1) = \mathrm{rank}(A_2) = 1$ and there exist unit vectors $u_1, u_2$ such that $A_1 = u_1 u_1^\top$, $A_2 = u_2 u_2^\top$, and $u_1 \neq \pm u_2$. Hence,

$$\mathrm{rank}(A_1 + A_2) = \mathrm{rank}([u_1, u_2][u_1, u_2]^\top) = \mathrm{rank}([u_1, u_2]) = 2.$$

Moreover, it follows from 48 that $A = \alpha_1(A_1 + A_2) + (\alpha_2 - \alpha_1)A_2$. Due to $\alpha_2 - \alpha_1 \geq 0$, one can get $\mathrm{rank}(A) \geq \mathrm{rank}(A_1 + A_2) = 2$ by a similar argument we applied in 49. However, this contradicts the assumption that $A$ is a rank-1 matrix. Therefore, for any unit vector $u$, $A = uu^\top$ is an extreme point of S. $\qquad\square$

**Lemma C.7.** *Suppose* $X \sim \mathcal{N}(0, \Sigma)$. *Then* $\mathbb{E}\|X\|_2 \geq \sqrt{\frac{2\operatorname{tr}(\Sigma)}{\pi}}$.

*Proof.* Without loss of generality, we can assume that $\operatorname{tr}(\Sigma) = 1$. Let $Z \sim \mathcal{N}(0, I)$. Since $\Sigma^{\frac{1}{2}}Z \sim \mathcal{N}(0, \Sigma)$, we have

$$\mathbb{E}\|X\|_2 = \mathbb{E}\|\Sigma^{\frac{1}{2}}Z\|_2 = \mathbb{E}\sqrt{Z^\top \Sigma Z}. \tag{50}$$

Define a function $f(A) := \mathbb{E}\sqrt{Z^\top A Z}$ and let $\mathcal{S}$ denote the set of all positive semidefinite matrices whose traces are equal to 1. Then $f(A)$ is continuous and concave over $\mathcal{S}$ that is convex and compact. By Bauer maximum principle, $f(A)$ attains its minimum at some extreme point $\widetilde{A}$ of $\mathcal{S}$. According to Lemma C.6, $\widetilde{A} = uu^\top$ with $\|u\|_2 = 1$. If follows that

$$\min_{A \in \mathcal{S}} f(A) = f(\widetilde{A}) = \mathbb{E}\sqrt{Z^\top \widetilde{A} Z} = \mathbb{E}|u^\top Z| = \sqrt{\frac{2}{\pi}}. \tag{51}$$

In the last step, we used the fact $u^\top Z \sim \mathcal{N}(0, 1)$. Combining 50 and 51, we obtain

$$\mathbb{E}\|X\|_2 = f(\Sigma) \geq \min_{A \in \mathcal{S}} f(A) = \sqrt{\frac{2}{\pi}}.$$

This completes the proof. $\qquad\square$

**Lemma C.8.** *Given an $n$-dimensional random vector $X \sim \mathcal{N}(0, \Sigma)$, we have*

$$\mathbb{E}\|\rho(X)\|_2 \geq \frac{1}{2}\mathbb{E}\|X\|_2$$

*where $\rho(x) = \max\{0, x\}$ is the ReLU activation function.*

*Proof.* We divide $\mathbb{R}^n$ into $J := 2^{n-1}$ pairs of orthants $\{(A_i, B_i)\}_{i=1}^J$ such that $-A_i = B_i$. For example, $\{(x_1, x_2, \ldots, x_n) : x_i > 0, i = 1, 2, \ldots, n\}$ and $\{(x_1, x_2, \ldots, x_n) : x_i < 0, i = 1, 2, \ldots, n\}$ compose one of these pairs. Since $X$ is symmetric, that is, $X$ and $-X$ have the same distribution, one can get

$$\int_{A_i} \|\rho(-x)\|_2 \, dP_X = \int_{B_i} \|\rho(x)\|_2 \, dP_X \tag{52}$$

and

$$\int_{A_i} \|x\|_2 \, dP_X = \int_{B_i} \|x\|_2 \, dP_X \tag{53}$$

where $P_X$ denotes the probability distribution of $X$. It follows that

$$\mathbb{E}\|\rho(X)\|_2 = \int_{\mathbb{R}^n} \|\rho(x)\|_2 \, dP_X$$

$$= \sum_{j=1}^J \int_{A_j \cup B_j} \|\rho(x)\|_2 \, dP_X$$

$$= \sum_{j=1}^J \int_{A_j} \|\rho(x)\|_2 \, dP_X + \int_{A_j} \|\rho(-x)\|_2 \, dP_X \qquad \text{(using 52)}$$

$$\geq \sum_{j=1}^J \int_{A_j} \|\rho(x) + \rho(-x)\|_2 \, dP_X \qquad \text{(by triangle inequality)}$$

$$= \sum_{j=1}^J \int_{A_j} \|x\|_2 \, dP_X$$

$$= \frac{1}{2}\sum_{j=1}^J \int_{A_j \cup B_j} \|x\|_2 \, dP_X \qquad \text{(using 53)}$$

$$= \frac{1}{2}\mathbb{E}\|X\|_2.$$

$\qquad\square$

Proposition C.5 then follows immediately from Lemma C.7 and Lemma C.8.

C.2   PERTURBATION ANALYSIS FOR UNDERDETERMINED SYSTEMS

In this section, we investigate the minimal $\ell_\infty$ norm solutions of perturbed underdetermined linear systems like 18, which can be used to bound the $\ell_\infty$ norm of $\widetilde{w}$ generated by the perfect data alignment. Specifically, consider a matrix $X \in \mathbb{R}^{m \times N}$ with $\mathrm{rank}(X) = m < N$. It admits the singular value decomposition

$$X = USV^\top \tag{54}$$

where $U = [U_1, \ldots, U_m] \in \mathbb{R}^{m \times m}$, $V = [V_1, \ldots, V_m] \in \mathbb{R}^{N \times m}$ have orthonormal columns, and $S = \mathrm{diag}(\sigma_1, \ldots, \sigma_m)$ consists of singular values $\sigma_1 \geq \sigma_2 \geq \ldots \geq \sigma_m > 0$. Moreover, suppose $\epsilon > 0$, $w \in \mathbb{R}^N$, and $E \in \mathbb{R}^{m \times N}$ satisfying $\|E\|_2 \leq \epsilon \|X\|_2$. Let $\widetilde{X} := X + E$ be the perturbed matrix and define

$$\hat{w} := \arg\min \|z\|_\infty \text{ subject to } Xz = Xw, \tag{55}$$

$$\widetilde{w} := \arg\min \|z\|_\infty \text{ subject to } \widetilde{X}z = Xw. \tag{56}$$

Our goal is to evaluate the ratio $\frac{\|\widetilde{w}\|_\infty}{\|\hat{w}\|_\infty}$.

The proposition below highlights the fact that one can construct systems where arbitrarily small perturbations can yield arbitrarily divergent solutions. The proof relies on the system being ill-conditioned, and on a particular construction of $X$ and $E$ to exploit the ill-conditioning.

**Proposition C.9.** *For $\epsilon, \gamma \in (0, 1)$, there exist a matrix $X \in \mathbb{R}^{m \times N}$, a perturbed version $\widetilde{X} = X + E$ with $\|E\|_2 \leq \epsilon \|X\|_2$, and a unit vector $w \in \mathbb{R}^N$, so that the optimal solutions to 55 and 56 satisfy $\frac{\|\widetilde{w}\|_\infty}{\|\hat{w}\|_\infty} = \frac{1}{\gamma}$.*

*Proof.* Let $U \in \mathbb{R}^{m \times m}$ be any orthogonal matrix and let $V \in \mathbb{R}^{N \times m}$ be the first $m$ columns of a normalized Hadamard matrix of order $N$. Then we have $V^\top V = I$ and entries of $V$ are either $\frac{1}{\sqrt{N}}$ or $-\frac{1}{\sqrt{N}}$. Set $X = USV^\top$ where $S \in \mathbb{R}^{m \times m}$ is diagonal with diagonal elements $\sigma_1 = \sigma_2 = \ldots = \sigma_{m-1} = 1$ and $\sigma_m = \epsilon$. Define a rank one matrix $E = \epsilon(\gamma - 1)U_m V_m^\top$. Then we have

$$\frac{\|E\|_2}{\|X\|_2} = \epsilon(1 - \gamma) < \epsilon, \quad \widetilde{X} = X + E = U\mathrm{diag}(1, \ldots, 1, \epsilon\gamma)V^\top.$$

Picking a unit vector $w = \epsilon V S^{-1} e_m$ with $e_m := (0, \ldots, 0, 1) \in \mathbb{R}^m$, the feasibility condition in 55, together with the definition of $X$, implies that $Xz = Xw$ is equivalent to

$$V^\top z = e_m. \tag{57}$$

Since $VV^\top z = P_{\mathrm{Im}(V)}(z)$ is the orthogonal projection of $z$ onto the image of $V$, for any feasible $z$ satisfying 57, we have

$$\|z\|_\infty \geq \frac{\|z\|_2}{\sqrt{N}} \geq \frac{\|VV^\top z\|_2}{\sqrt{N}} = \frac{\|V^\top z\|_2}{\sqrt{N}} = \frac{\|e_m\|_2}{\sqrt{N}} = \frac{1}{\sqrt{N}}.$$

Note that $z = V_m$ satisfies 57 and $\|V_m\|_\infty = \frac{1}{\sqrt{N}}$ achieves the lower bound. Thus, we have found an optimal solution $\hat{w} = V_m$ with $\|\hat{w}\|_\infty = \frac{1}{\sqrt{N}}$.

Meanwhile the corresponding feasibility condition in 56, coupled with the definition of $\widetilde{X}$, implies that $\widetilde{X}z = Xw$ can be rewritten as $V^\top z = \frac{1}{\gamma} e_m$. By a similar argument we used for solving 55, we obtain that $\widetilde{w} = \frac{1}{\gamma} V_m$ is an optimal solution to 56 and thus $\|\widetilde{w}\|_\infty = \frac{1}{\gamma \sqrt{N}}$. Therefore, we have $\frac{\|\widetilde{w}\|_\infty}{\|\hat{w}\|_\infty} = \frac{1}{\gamma}$ as desired. $\qquad\square$

Proposition C.9 constructs a scenario in which adjusting the weights to achieve $\widetilde{X}\widetilde{w} = X\hat{w} = Xw$, under even a small perturbation of $X$, inexorably leads to a large increase in the infinity norm of $\widetilde{w}$. In Proposition C.11, we consider a more reasonable scenario where the original weights $w$ is Gaussian that is more likely to be representative of ones encountered in practice. The proof of the following lemma follows (, https://mathoverflow.net/users/36721/iosif pinelis).

**Lemma C.10.** *Let $\| \cdot \|$ be any vector norm on $\mathbb{R}^n$. Let $X \sim \mathcal{N}(0, \Sigma_1)$ and $Y \sim \mathcal{N}(0, \Sigma_2)$ be $n$-dimensional random vectors. Suppose $\Sigma_1 \preceq \Sigma_2$. Then, for $t \geq 0$, we have*

$$\mathrm{P}(\|X\| \leq t) \geq \mathrm{P}(\|Y\| \leq t).$$

*Proof.* Fix $t \geq 0$. Define $g : \mathbb{R}^n \to [0, 1]$ by

$$g(z) := \mathrm{P}(\|X + z\| \leq t) = \int_{\mathbb{R}^n} f_X(x) \mathbb{1}_{\{\|x+z\| \leq t\}} \, dx$$

where $f_X(x) := (2\pi)^{-\frac{n}{2}} \det(\Sigma_1)^{-\frac{1}{2}} \exp(-\frac{1}{2} x^\top \Sigma_1^{-1} x)$ is the density function of $X$. Since $\log f_X(x) = -\frac{1}{2} x^\top \Sigma_1^{-1} x$ is concave and $\mathbb{1}_{\{\|x+z\| \leq t\}}$ is an indicator function of a convex set, both $f_X(x)$ and $\mathbb{1}_{\{\|x+z\| \leq t\}}$ are log-concave. It follows that the product $h(x, z) := f_X(x) \mathbb{1}_{\{\|x+z\| \leq t\}}$ is also log-concave. Applying the Prékopa–Leindler inequality (Prékopa, 1971; 1973), the marginalization $g(z) = \int_{\mathbb{R}^n} h(x, z) \, dx$ preserves log-concavity. Additionally, by change of variables and the symmetry of $f_X(x)$, we have

$$g(-z) = \int_{\mathbb{R}^n} f_X(x) \mathbb{1}_{\{\|x-z\| \leq t\}} \, dx = \int_{\mathbb{R}^n} f_X(x) \mathbb{1}_{\{\|x+z\| \leq t\}} \, dx = g(z).$$

So $g(z)$ is a log-concave even function, which implies that, for any $z \in \mathbb{R}^n$,

$$g(z) = g(z)^{\frac{1}{2}} g(-z)^{\frac{1}{2}} \leq g\left(\frac{1}{2} z - \frac{1}{2} z\right) = g(0) = \mathrm{P}(\|X\| \leq t). \tag{58}$$

Now, let $Z \sim \mathcal{N}(0, \Sigma_2 - \Sigma_1)$ be independent of $X$. Then $X + Z \overset{d}{=} Y \sim \mathcal{N}(0, \Sigma_2)$ and, by 58, $\mathbb{E}g(Z) \leq \mathrm{P}(\|X\| \leq t)$. It follows that

$$
\begin{aligned}
\mathrm{P}(\|X\| \leq t) &\geq \mathbb{E}g(Z) \\
&= \int_{\mathbb{R}^n} f_Z(z) g(z) \, dz \\
&= \int_{\mathbb{R}^n} \int_{\mathbb{R}^n} f_X(x) f_Z(z) \mathbb{1}_{\{\|x+z\| \leq t\}} \, dx \, dz \\
&= \int_{\mathbb{R}^n \times \mathbb{R}^n} f_{(X,Z)}(x, z) \mathbb{1}_{\{\|x+z\| \leq t\}} \, d(x, z) \\
&= \mathrm{P}(\|X + Z\| \leq t) \\
&= \mathrm{P}(\|Y\| \leq t)
\end{aligned}
$$

where $f_Z(z)$ and $f_{(X,Z)}(x, z)$ are density functions of $Z$ and $(X, Z)$ respectively. $\square$

**Proposition C.11.** *Let $X \in \mathbb{R}^{m \times N}$ admit the singular value decomposition $X = USV^\top$ as in 54 and let $w \in \mathbb{R}^N$ be a random vector with i.i.d. $\mathcal{N}(0, 1)$ entries. Let $p \in \mathbb{N}$ with $p \geq 2$. Given $\epsilon \in (0, 1)$, suppose $\widetilde{X} = X + E \in \mathbb{R}^{m \times N}$ with $\|E\|_2 \leq \epsilon \sigma_1 < \sigma_m$. Then, with probability at least $1 - \frac{2}{N^{p-1}}$,*

$$\|\widetilde{w}\|_\infty \leq \frac{\sigma_1}{\sigma_m - \epsilon \sigma_1} \sqrt{2p \log N}$$

*holds for all optimal solutions $\widetilde{w}$ of 56.*

*Proof.* Let $w^\sharp := VV^\top w$ be the orthogonal projection of $w$ onto $\mathrm{Im}(V)$. Let $\widetilde{V} = [V, \hat{V}] \in \mathbb{R}^{N \times N}$ be an expansion of $V$ such that $\widetilde{V}$ is orthogonal. Define

$$\widetilde{\mathcal{E}} := U^\top E \widetilde{V} = [U^\top EV, U^\top E\hat{V}] = [\mathcal{E}, \hat{\mathcal{E}}] \in \mathbb{R}^{m \times N}$$

where $\mathcal{E} := U^\top EV$ and $\hat{\mathcal{E}} := U^\top E\hat{V}$. Then $E = U\widetilde{\mathcal{E}}\widetilde{V}^\top$ and thus

$$\epsilon \sigma_1 \geq \|E\|_2 = \|\widetilde{\mathcal{E}}\|_2 \geq \|\mathcal{E}\|_2. \tag{59}$$

Define $z^\sharp := V (S + \mathcal{E})^{-1} S V^\top w \in \mathbb{R}^N$. Since $\widetilde{\mathcal{E}} \widetilde{V}^\top V = \mathcal{E}$, we have

$$
\begin{aligned}
\widetilde{X} z^\sharp &= X z^\sharp + E z^\sharp \\
&= U S (S + \mathcal{E})^{-1} S V^\top w + U \widetilde{\mathcal{E}} \widetilde{V}^\top V (S + \mathcal{E})^{-1} S V^\top w \\
&= U S (S + \mathcal{E})^{-1} S V^\top w + U \mathcal{E} (S + \mathcal{E})^{-1} S V^\top w \\
&= U S V^\top w \\
&= X w.
\end{aligned}
$$

Moreover, since $w \sim \mathcal{N}(0, I)$, we have $z^\sharp \sim \mathcal{N}(0, BB^\top)$ with $B := V(S + \mathcal{E})^{-1} S$ and thus

$$
BB^\top \preccurlyeq \|BB^\top\|_2 I = \|B\|_2^2 I = \|(S + \mathcal{E})^{-1} S\|_2^2 I \preccurlyeq \left( \frac{\sigma_1}{\sigma_m - \|\mathcal{E}\|_2} \right)^2 I \preccurlyeq \left( \frac{\sigma_1}{\sigma_m - \epsilon \sigma_1} \right)^2 I. \quad (60)
$$

Applying Lemma C.10 to 60 with $\Sigma_1 = BB^\top$ and $\Sigma_2 = (\frac{\sigma_1}{\sigma_m - \epsilon \sigma_1})^2 I$, we obtain that, for $t \geq 0$,

$$
\mathrm{P}(\|z^\sharp\|_\infty \leq t) \geq \mathrm{P}\left( \left\| \frac{\sigma_1 \xi}{\sigma_m - \epsilon \sigma_1} \right\|_\infty \leq t \right) \geq 1 - 2N \exp\left( -\frac{1}{2} \left( \frac{\sigma_m - \epsilon \sigma_1}{\sigma_1} \right)^2 t^2 \right) \quad (61)
$$

where $\xi \sim \mathcal{N}(0, I)$. In the last inequality, we used the following concentration inequality

$$
\mathrm{P}(\|\xi\|_\infty \leq t) \geq 1 - 2N e^{-\frac{t^2}{2}}, \quad t \geq 0.
$$

Choosing $t = \frac{\sigma_1}{\sigma_m - \epsilon \sigma_1} \sqrt{2p \log N}$ in 61, we obtain

$$
\mathrm{P}\left( \|z^\sharp\|_\infty \leq \frac{\sigma_1}{\sigma_m - \epsilon \sigma_1} \sqrt{2p \log N} \right) \geq 1 - \frac{2}{N^{p-1}}.
$$

Further, since $z^\sharp$ is a feasible vector of 56, we have $\|\widetilde{w}\|_\infty \leq \|z^\sharp\|_\infty$. Therefore, with probability at least $1 - \frac{2}{N^{p-1}}$,

$$
\|\widetilde{w}\|_\infty \leq \frac{\sigma_1}{\sigma_m - \epsilon \sigma_1} \sqrt{2p \log N}.
$$

$\square$

# D  PROOF OF THEOREM 3.1

In this section, we will prove Theorem 3.1 using a sequence of results which we will now briefly outline. Recall here that on the one hand, in Algorithm 1 with perfect data alignment, since data is aligned by solving 18, we only have to bound the quantization error $\widetilde{u}_{N_{i-1}}$ generated by procedure 15. On the other hand, Algorithm 1 with approximate data alignment has a faster implementation provided $r < m$, but introduces an extra error $\hat{u}_{r N_{i-1}}$ arising from the $r$-th order data alignment. Thus, to control the error bounds, we first bound $\widetilde{u}_{N_{i-1}}$ and $\hat{u}_{r N_{i-1}}$ appearing in 19 and 21 in Lemma D.1 and Lemma D.2 respectively. With these bounds in hand, we next prove Theorem D.3 which provides a recursive relation between the error in the current layer and that of the previous layer. Finally, applying Theorem D.3 inductively over all layers, we complete the proof.

**Lemma D.1** (Quantization error). *Assuming that the first $i - 1$ layers have been quantized, let $X^{(i-1)}, \widetilde{X}^{(i-1)}$ be as in 7 and $w \in \mathbb{R}^{N_{i-1}}$ be the weights associated with a neuron in the $i$-th layer, i.e. a column of $W^{(i)} \in \mathbb{R}^{N_{i-1} \times N_i}$. Suppose $\widetilde{w}$ is either the solution of 18 or the output of 20. Quantize $\widetilde{w}$ using 15 with alphabets $\mathcal{A} = \mathcal{A}_\infty^\delta$ as in 4. Then, for any $p \in \mathbb{N}$,*

$$
\|\widetilde{u}_{N_{i-1}}\|_2 \leq \delta \sqrt{2\pi p m \log N_{i-1}} \max_{1 \leq j \leq N_{i-1}} \|\widetilde{X}_j^{(i-1)}\|_2 \quad (62)
$$

*holds with probability at least $1 - \frac{\sqrt{2} m}{N_{i-1}^p}$.*

*Proof.* We first show that

$$
\widetilde{u}_t \leq_{\mathrm{cx}} \mathcal{N}(0, \Sigma_t) \quad (63)
$$

holds for all $1 \leq t \leq N_{i-1}$, where $\Sigma_t$ is defined recursively as follows

$$
\Sigma_t := P_{\widetilde{X}_t^{(i-1)\perp}} \Sigma_{t-1} P_{\widetilde{X}_t^{(i-1)\perp}} + \frac{\pi \delta^2}{2} \widetilde{X}_t^{(i-1)} \widetilde{X}_t^{(i-1)\top} \quad \text{with} \quad \Sigma_0 := 0.
$$

At the $t$-th step of quantizing $\widetilde{w}$, by 15, we have $\widetilde{u}_t = \widetilde{u}_{t-1} + (\widetilde{w}_t - \widetilde{q}_t)\widetilde{X}_t^{(i-1)}$. Define

$$h_t := \widetilde{u}_{t-1} + \widetilde{w}_t \widetilde{X}_t^{(i-1)} \quad \text{and} \quad v_t := \frac{\langle \widetilde{X}_t^{(i-1)}, h_t \rangle}{\|\widetilde{X}_t^{(i-1)}\|_2^2}. \tag{64}$$

It follows that

$$\widetilde{u}_t = h_t - \widetilde{q}_t \widetilde{X}_t^{(i-1)} \tag{65}$$

and 15 implies

$$\widetilde{q}_t = \mathcal{Q}_{\text{StocQ}}\left( \frac{\langle \widetilde{X}_t^{(i-1)}, h_t \rangle}{\|\widetilde{X}_t^{(i-1)}\|_2^2} \right) = \mathcal{Q}_{\text{StocQ}}(v_t). \tag{66}$$

Since $\mathcal{A} = \mathcal{A}_\infty^\delta$, $\mathbb{E}\mathcal{Q}_{\text{StocQ}}(z) = z$ for all $z \in \mathbb{R}$. Moreover, conditioning on $\widetilde{u}_{t-1}$ in 64, $h_t$ and $v_t$ are fixed and thus one can get

$$\mathbb{E}(\mathcal{Q}_{\text{StocQ}}(v_t)|\widetilde{u}_{t-1}) = v_t \tag{67}$$

and

$$\begin{aligned}
\mathbb{E}(\widetilde{u}_t|\widetilde{u}_{t-1}) &= \mathbb{E}(h_t - \widetilde{q}_t \widetilde{X}_t^{(i-1)}|\widetilde{u}_{t-1}) \\
&= h_t - \widetilde{X}_t^{(i-1)}\mathbb{E}(\widetilde{q}_t|\widetilde{u}_{t-1}) \\
&= h_t - \widetilde{X}_t^{(i-1)}\mathbb{E}(\mathcal{Q}_{\text{StocQ}}(v_t)|\widetilde{u}_{t-1}) \\
&= h_t - v_t \widetilde{X}_t^{(i-1)} \\
&= h_t - \frac{\langle \widetilde{X}_t^{(i-1)}, h_t \rangle}{\|\widetilde{X}_t^{(i-1)}\|_2^2} \widetilde{X}_t^{(i-1)} \\
&= \left( I - \frac{\widetilde{X}_t^{(i-1)} \widetilde{X}_t^{(i-1)\top}}{\|\widetilde{X}_t^{(i-1)}\|_2^2} \right) h_t \\
&= P_{\widetilde{X}_t^{(i-1)\perp}}(h_t).
\end{aligned}$$

The identity above indicates that the approximation error $\widetilde{u}_t$ can be split into two parts: its conditional mean $P_{\widetilde{X}_t^{(i-1)\perp}}(h_t)$ and a random perturbation. Specifically, applying 65 and 35, we obtain

$$\widetilde{u}_t = P_{\widetilde{X}_t^{(i-1)\perp}}(h_t) + P_{\widetilde{X}_t^{(i-1)}}(h_t) - \widetilde{q}_t \widetilde{X}_t^{(i-1)} = P_{\widetilde{X}_t^{(i-1)\perp}}(h_t) + R_t \widetilde{X}_t^{(i-1)} \tag{68}$$

where

$$R_t := v_t - \widetilde{q}_t.$$

Further, combining 66 and 67, we have

$$\mathbb{E}(R_t|\widetilde{u}_{t-1}) = v_t - \mathbb{E}(\widetilde{q}_t|\widetilde{u}_{t-1}) = v_t - \mathbb{E}(\mathcal{Q}_{\text{StocQ}}(v_t)|\widetilde{u}_{t-1}) = 0$$

and $|R_t| = |v_t - \mathcal{Q}_{\text{StocQ}}(v_t)| \leq \delta$. Lemma A.6 yields that, conditioning on $\widetilde{u}_{t-1}$,

$$R_t \leq_{\text{cx}} \mathcal{N}\left(0, \frac{\pi\delta^2}{2}\right). \tag{69}$$

Now, we are ready to prove 63 by induction on $t$. When $t = 1$, we have $h_1 = \widetilde{w}_1 \widetilde{X}_1^{(i-1)}$. We can deduce from 68 and 69 that $\widetilde{u}_1 = P_{\widetilde{X}_1^{(i-1)\perp}}(\widetilde{w}_1 \widetilde{X}_1^{(i-1)}) + R_1 \widetilde{X}_1^{(i-1)} = R_1 \widetilde{X}_1^{(i-1)}$ with $R_1 \leq_{\text{cx}} \mathcal{N}\left(0, \frac{\pi\delta^2}{2}\right)$. Applying Lemma A.4, we obtain $\widetilde{u}_1 \leq_{\text{cx}} \mathcal{N}(0, \Sigma_1)$. Next, assume that 63 holds for $t - 1$ with $t \geq 2$. By the induction hypothesis, we have $\widetilde{u}_{t-1} \leq_{\text{cx}} \mathcal{N}(0, \Sigma_{t-1})$. Using Lemma A.4 again, we get

$$\begin{aligned}
P_{\widetilde{X}_t^{(i-1)\perp}}(h_t) &= P_{\widetilde{X}_t^{(i-1)\perp}}(\widetilde{u}_{t-1} + \widetilde{w}_t \widetilde{X}_t^{(i-1)}) \\
&\leq_{\text{cx}} \mathcal{N}\left( P_{\widetilde{X}_t^{(i-1)\perp}}(\widetilde{w}_t \widetilde{X}_t^{(i-1)}), P_{\widetilde{X}_t^{(i-1)\perp}}\Sigma_{t-1}P_{\widetilde{X}_t^{(i-1)\perp}} \right) \\
&= \mathcal{N}\left( 0, P_{\widetilde{X}_t^{(i-1)\perp}}\Sigma_{t-1}P_{\widetilde{X}_t^{(i-1)\perp}} \right).
\end{aligned}$$

Additionally, conditioning on $\widetilde{u}_{t-1}$, 69 implies

$$R_t \widetilde{X}_t^{(i-1)} \leq_{\mathrm{cx}} \mathcal{N}\Big(0, \frac{\pi\delta^2}{2} \widetilde{X}_t^{(i-1)} \widetilde{X}_t^{(i-1)\top}\Big).$$

Then we apply Lemma A.5 to 68 by taking

$$X = P_{\widetilde{X}_t^{(i-1)\perp}}(h_t), \ Y = \widetilde{u}_t, \ W = \mathcal{N}\Big(0, P_{\widetilde{X}_t^{(i-1)\perp}} \Sigma_{t-1} P_{\widetilde{X}_t^{(i-1)\perp}}\Big), \ Z = \mathcal{N}\Big(0, \frac{\pi\delta^2}{2} \widetilde{X}_t^{(i-1)} \widetilde{X}_t^{(i-1)\top}\Big).$$

It follows that

$$\begin{aligned}
\widetilde{u}_t &\leq_{\mathrm{cx}} W + Z \\
&= \mathcal{N}\Big(0, P_{\widetilde{X}_t^{(i-1)\perp}} \Sigma_{t-1} P_{\widetilde{X}_t^{(i-1)\perp}} + \frac{\pi\delta^2}{2} \widetilde{X}_t^{(i-1)} \widetilde{X}_t^{(i-1)\top}\Big) \\
&= \mathcal{N}(0, \Sigma_t).
\end{aligned}$$

Here, we used the independence of $W$ and $Z$, and the definition of $\Sigma_t$. This establishes inequality 63 showing that $\widetilde{u}_t$ is dominated by $\mathcal{N}(0, \Sigma_t)$ in the convex order, where $\Sigma_t$ is defined recursively using orthogonal projections. So it remains to control the covariance matrix $\Sigma_t$. Recall that $\Sigma_t$ defined as follows.

$$\Sigma_t = P_{\widetilde{X}_t^{(i-1)\perp}} \Sigma_{t-1} P_{\widetilde{X}_t^{(i-1)\perp}} + \frac{\pi\delta^2}{2} \widetilde{X}_t^{(i-1)} \widetilde{X}_t^{(i-1)\top} \quad \text{with} \quad \Sigma_0 = 0.$$

Then we apply Lemma C.1 with $M_t = \Sigma_t$, $z_t = \widetilde{X}_t^{(i-1)}$, and $\alpha = \frac{\pi\delta^2}{2}$, and conclude that $\Sigma_t \preceq \sigma_t^2 I$ with $\sigma_t^2 = \frac{\pi\delta^2}{2} \max_{1 \leq j \leq t} \|\widetilde{X}_j^{(i-1)}\|_2^2$. Note that $\widetilde{u}_t \leq_{\mathrm{cx}} \mathcal{N}(0, \Sigma_t)$ and, by Lemma A.3, we have $\mathcal{N}(0, \Sigma_t) \leq_{\mathrm{cx}} \mathcal{N}(0, \sigma_t^2 I)$. Then we deduce from the transitivity of $\leq_{\mathrm{cx}}$ that $\widetilde{u}_t \leq_{\mathrm{cx}} \mathcal{N}(0, \sigma_t^2 I)$. It follows from Lemma C.2 that, for $\gamma \in (0, 1]$ and $1 \leq t \leq N_{i-1}$,

$$\mathrm{P}\Big( \|\widetilde{u}_t\|_\infty \leq 2\sigma_t \sqrt{\log(\sqrt{2}m/\gamma)} \Big) \geq 1 - \gamma.$$

Picking $\gamma = \sqrt{2}m N_{i-1}^{-p}$ and $t = N_{i-1}$,

$$\|\widetilde{u}_{N_{i-1}}\|_2 \leq \sqrt{m}\|\widetilde{u}_{N_{i-1}}\|_\infty \leq 2\sigma_{N_{i-1}}\sqrt{pm \log N_{i-1}} = \delta\sqrt{2\pi pm \log N_{i-1}} \max_{1 \leq j \leq N_{i-1}} \|\widetilde{X}_j^{(i-1)}\|_2$$

holds with probability exceeding $1 - \sqrt{2}m N_{i-1}^{-p}$. $\qquad\square$

Next, we deduce a closed-form expression of $\hat{u}_{rN_{i-1}}$ showing that $\|\hat{u}_{rN_{i-1}}\|_2$ decays polynomially with respect to $r$.

**Lemma D.2** (Data alignment error). *Assuming that the first $i-1$ layers have been quantized, let $X^{(i-1)}, \widetilde{X}^{(i-1)}$ be as in 7 and let $w \in \mathbb{R}^{N_{i-1}}$ be a neuron in the $i$-th layer, i.e. a column of $W^{(i)} \in \mathbb{R}^{N_{i-1} \times N_i}$. Applying the $r$-th order data alignment procedure in 13 and 20, we have*

$$\hat{u}_{N_{i-1}} = \sum_{j=1}^{N_{i-1}} w_j P_{\widetilde{X}_{N_{i-1}}^{(i-1)\perp}} \dots P_{\widetilde{X}_{j+1}^{(i-1)\perp}} P_{\widetilde{X}_j^{(i-1)\perp}} (X_j^{(i-1)}) \tag{70}$$

*and*

$$\hat{u}_{rN_{i-1}} = (P^{(i-1)})^{r-1} \hat{u}_{N_{i-1}} \tag{71}$$

*where $P^{(i-1)} := P_{\widetilde{X}_{N_{i-1}}^{(i-1)\perp}} \dots P_{\widetilde{X}_2^{(i-1)\perp}} P_{\widetilde{X}_1^{(i-1)\perp}}$.*

*Proof.* We first prove the following identity by induction on $t$.

$$\hat{u}_t = \sum_{j=1}^{t} w_j P_{\widetilde{X}_t^{(i-1)\perp}} \dots P_{\widetilde{X}_{j+1}^{(i-1)\perp}} P_{\widetilde{X}_j^{(i-1)\perp}} (X_j^{(i-1)}), \quad 1 \leq t \leq N_{i-1}. \tag{72}$$

By 13, the case $t = 1$ is straightforward, since we have

$$\hat{u}_1 = w_1 X_1^{(i-1)} - \widetilde{w}_1 \widetilde{X}_1^{(i-1)}$$

$$= w_1 X_1^{(i-1)} - \frac{\langle \widetilde{X}_1^{(i-1)}, w_1 X_1^{(i-1)} \rangle}{\|\widetilde{X}_1^{(i-1)}\|_2^2} \widetilde{X}_1^{(i-1)}$$

$$= w_1 X_1^{(i-1)} - P_{\widetilde{X}_1^{(i-1)}}(w_1 X_1^{(i-1)})$$

$$= w_1 P_{\widetilde{X}_1^{(i-1)\perp}}(X_1^{(i-1)})$$

where we apply the properties of orthogonal projections in 35 and 36. For $2 \le t \le N_{i-1}$, assume that 72 holds for $t-1$. Then, by 13, one gets

$$\hat{u}_t = \hat{u}_{t-1} + w_t X_t^{(i-1)} - \widetilde{w}_t \widetilde{X}_t^{(i-1)}$$

$$= \hat{u}_{t-1} + w_t X_t^{(i-1)} - \frac{\langle \widetilde{X}_t^{(i-1)}, \hat{u}_{t-1} + w_t X_t^{(i-1)} \rangle}{\|\widetilde{X}_t^{(i-1)}\|_2^2} \widetilde{X}_t^{(i-1)}$$

$$= \hat{u}_{t-1} + w_t X_t^{(i-1)} - P_{\widetilde{X}_t^{(i-1)}}(\hat{u}_{t-1} + w_t X_t^{(i-1)})$$

$$= P_{\widetilde{X}_t^{(i-1)\perp}}(\hat{u}_{t-1} + w_t X_t^{(i-1)}).$$

Applying the induction hypothesis, we obtain

$$\hat{u}_t = P_{\widetilde{X}_t^{(i-1)\perp}}(\hat{u}_{t-1}) + w_t P_{\widetilde{X}_t^{(i-1)\perp}}(X_t^{(i-1)})$$

$$= \sum_{j=1}^{t-1} w_j P_{\widetilde{X}_t^{(i-1)\perp}} \dots P_{\widetilde{X}_{j+1}^{(i-1)\perp}} P_{\widetilde{X}_j^{(i-1)\perp}}(X_j^{(i-1)}) + w_t P_{\widetilde{X}_t^{(i-1)\perp}}(X_t^{(i-1)})$$

$$= \sum_{j=1}^{t} w_j P_{\widetilde{X}_t^{(i-1)\perp}} \dots P_{\widetilde{X}_{j+1}^{(i-1)\perp}} P_{\widetilde{X}_j^{(i-1)\perp}}(X_j^{(i-1)}).$$

This completes the proof of 72. In particular, if $t = N_{i-1}$, then we obtain 70.

Next, we consider $\hat{u}_t$ when $t > N_{i-1}$. Plugging $t = N_{i-1} + 1$ into 20, and recalling that our indices (except for $\hat{u}$) are modulo $N_{i-1}$, we have

$$\hat{u}_{N_{i-1}+1} = \hat{u}_{N_{i-1}} + \widetilde{w}_1 \widetilde{X}_1^{(i-1)} - \frac{\langle \widetilde{X}_1^{(i-1)}, \hat{u}_{N_{i-1}} + \widetilde{w}_1 \widetilde{X}_1^{(i-1)} \rangle}{\|\widetilde{X}_1^{(i-1)}\|_2^2} \widetilde{X}_1^{(i-1)} = P_{\widetilde{X}_1^{(i-1)\perp}}(\hat{u}_{N_{i-1}}).$$

Similarly, one can show that $\hat{u}_{N_{i-1}+2} = P_{\widetilde{X}_2^{(i-1)\perp}}(\hat{u}_{N_{i-1}+1}) = P_{\widetilde{X}_2^{(i-1)\perp}} P_{\widetilde{X}_1^{(i-1)\perp}} \hat{u}_{N_{i-1}}$. Repeating this argument for all $N_{i-1} < t \le r N_{i-1}$, we can derive 71. □

Combining Lemma D.1 and Lemma D.2, we can derive a recursive relation between the error in the current layer and that of the previous layer.

**Theorem D.3.** *Let $\Phi$ be an $L$-layer neural network as in 3 where the activation function is $\varphi^{(i)}(x) = \rho(x) := \max\{0, x\}$ for $1 \le i \le L$. Let $\mathcal{A} = \mathcal{A}_\infty^\delta$ be as in 4 and $p \in \mathbb{N}$.*

*(a) If we quantize $\Phi$ using Algorithm 1 with perfect data alignment, then, for each $2 \le i \le L$,*

$$\max_{1 \le j \le N_i} \|X^{(i-1)} W_j^{(i)} - \widetilde{X}^{(i-1)} Q_j^{(i)}\|_2 \le \delta \sqrt{2\pi pm \log N_{i-1}} \max_{1 \le j \le N_{i-1}} \|X_j^{(i-1)}\|_2$$

$$+ \delta \sqrt{2\pi pm \log N_{i-1}} \max_{1 \le j \le N_{i-1}} \|X^{(i-2)} W_j^{(i-1)} - \widetilde{X}^{(i-2)} Q_j^{(i-1)}\|_2.$$

*holds with probability at least $1 - \frac{\sqrt{2} m N_i}{N_{i-1}^p}$.*

*(b) If we quantize $\Phi$ using Algorithm 1 with approximate data alignment, then, for each $2 \le i \le L$,*

$$\max_{1 \le j \le N_i} \|X^{(i-1)} W_j^{(i)} - \widetilde{X}^{(i-1)} Q_j^{(i)}\|_2 \le \delta \sqrt{2\pi pm \log N_{i-1}} \max_{1 \le j \le N_{i-1}} \|X_j^{(i-1)}\|_2$$

$$+ \left( N_{i-1} \|W^{(i)}\|_{\max} \|P^{(i-1)}\|_2^{r-1} + \delta \sqrt{2\pi pm \log N_{i-1}} \right) \max_{1 \le j \le N_{i-1}} \|X^{(i-2)} W_j^{(i-1)} - \widetilde{X}^{(i-2)} Q_j^{(i-1)}\|_2$$

*holds with probability exceeding* $1 - \frac{\sqrt{2}mN_i}{N_{i-1}^p}$. *Here,* $P^{(i-1)}$ *is defined in Lemma D.2.*

*Proof.* $(a)$ Note that, for each $1 \leq j \leq N_i$, the $j$-th columns $W_j^{(i)}$ and $Q_j^{(i)}$ represent a neuron and its quantized version respectively. Applying 19 and 62, we obtain

$$\mathrm{P}\Big(\|X^{(i-1)}W_j^{(i)} - \widetilde{X}^{(i-1)}Q_j^{(i)}\|_2 \leq \delta\sqrt{2\pi pm\log N_{i-1}} \max_{1\leq j\leq N_{i-1}} \|\widetilde{X}_j^{(i-1)}\|_2\Big) \geq 1 - \frac{\sqrt{2}m}{N_{i-1}^p}.$$

Taking a union bound over all $j$,

$$\max_{1\leq j\leq N_i} \|X^{(i-1)}W_j^{(i)} - \widetilde{X}^{(i-1)}Q_j^{(i)}\|_2 \leq \delta\sqrt{2\pi pm\log N_{i-1}} \max_{1\leq j\leq N_{i-1}} \|\widetilde{X}_j^{(i-1)}\|_2$$

holds with probability at least $1 - \frac{\sqrt{2}mN_i}{N_{i-1}^p}$. By the triangle inequality, we have

$$\begin{aligned}
\max_{1\leq j\leq N_{i-1}} \|\widetilde{X}_j^{(i-1)}\|_2 &\leq \max_{1\leq j\leq N_{i-1}} \|X_j^{(i-1)}\|_2 + \max_{1\leq j\leq N_{i-1}} \|X_j^{(i-1)} - \widetilde{X}_j^{(i-1)}\|_2 \\
&= \max_{1\leq j\leq N_{i-1}} \|X_j^{(i-1)}\|_2 + \max_{1\leq j\leq N_{i-1}} \|\rho(X^{(i-2)}W_j^{(i-1)}) - \rho(\widetilde{X}^{(i-2)}Q_j^{(i-1)})\|_2 \\
&\leq \max_{1\leq j\leq N_{i-1}} \|X_j^{(i-1)}\|_2 + \max_{1\leq j\leq N_{i-1}} \|X^{(i-2)}W_j^{(i-1)} - \widetilde{X}^{(i-2)}Q_j^{(i-1)}\|_2
\end{aligned} \tag{73}$$

It follows that, with probability at least $1 - \frac{\sqrt{2}mN_i}{N_{i-1}^p}$,

$$\begin{aligned}
\max_{1\leq j\leq N_i} \|X^{(i-1)}W_j^{(i)} - \widetilde{X}^{(i-1)}Q_j^{(i)}\|_2 &\leq \delta\sqrt{2\pi pm\log N_{i-1}} \max_{1\leq j\leq N_{i-1}} \|X_j^{(i-1)}\|_2 \\
&+ \delta\sqrt{2\pi pm\log N_{i-1}} \max_{1\leq j\leq N_{i-1}} \|X^{(i-2)}W_j^{(i-1)} - \widetilde{X}^{(i-2)}Q_j^{(i-1)}\|_2.
\end{aligned}$$

$(b)$ Applying Lemma D.2 with $w = W_j^{(i)}$ and using the fact that $\|P\|_2 \leq 1$ for any orthogonal projection $P$, we have

$$\begin{aligned}
\|\hat{u}_{N_{i-1}}\|_2 &= \Big\| \sum_{k=1}^{N_{i-1}} W_{kj}^{(i)} P_{\widetilde{X}_{N_{i-1}}^{(i-1)\perp}} \ldots P_{\widetilde{X}_{k+1}^{(i-1)\perp}} P_{\widetilde{X}_k^{(i-1)\perp}}(X_k^{(i-1)}) \Big\|_2 \\
&\leq \sum_{k=1}^{N_{i-1}} |W_{kj}^{(i)}| \Big\| P_{\widetilde{X}_k^{(i-1)\perp}}(X_k^{(i-1)}) \Big\|_2 \\
&= \sum_{k=1}^{N_{i-1}} |W_{kj}^{(i)}| \Big\| P_{\widetilde{X}_k^{(i-1)\perp}}(X_k^{(i-1)} - \widetilde{X}_k^{(i-1)}) \Big\|_2 \\
&\leq N_{i-1} \|W_j^{(i)}\|_\infty \max_{1\leq j\leq N_{i-1}} \|X_j^{(i-1)} - \widetilde{X}_j^{(i-1)}\|_2 \\
&= N_{i-1} \|W_j^{(i)}\|_\infty \max_{1\leq j\leq N_{i-1}} \|\rho(X^{(i-2)}W_j^{(i-1)}) - \rho(\widetilde{X}^{(i-2)}Q_j^{(i-1)})\|_2 \\
&\leq N_{i-1} \|W^{(i)}\|_{\max} \max_{1\leq j\leq N_{i-1}} \|X^{(i-2)}W_j^{(i-1)} - \widetilde{X}^{(i-2)}Q_j^{(i-1)}\|_2.
\end{aligned} \tag{74}$$

Then it follows from 21, 62, 73, and 74 that

$$\begin{aligned}
&\|X^{(i-1)}W_j^{(i)} - \widetilde{X}^{(i-1)}Q_j^{(i)}\|_2 \\
&\leq \|\hat{u}_{rN_{i-1}}\|_2 + \|\tilde{u}_{N_{i-1}}\|_2 \\
&\leq \|P^{(i-1)}\|_2^{r-1} \|\hat{u}_{N_{i-1}}\|_2 + \delta\sqrt{2\pi pm\log N_{i-1}} \max_{1\leq j\leq N_{i-1}} \|\widetilde{X}_j^{(i-1)}\|_2 \\
&\leq N_{i-1} \|W^{(i)}\|_{\max} \|P^{(i-1)}\|_2^{r-1} \max_{1\leq j\leq N_{i-1}} \|X^{(i-2)}W_j^{(i-1)} - \widetilde{X}^{(i-2)}Q_j^{(i-1)}\|_2 + \delta\sqrt{2\pi pm\log N_{i-1}} \\
&\times \Big( \max_{1\leq j\leq N_{i-1}} \|X_j^{(i-1)}\|_2 + \max_{1\leq j\leq N_{i-1}} \|X^{(i-2)}W_j^{(i-1)} - \widetilde{X}^{(i-2)}Q_j^{(i-1)}\|_2 \Big)
\end{aligned}$$

holds with probability at least $1 - \sqrt{2}mN_{i-1}^{-p}$. By a union bound over all $j$, we obtain that

$$\max_{1 \le j \le N_i} \|X^{(i-1)}W_j^{(i)} - \widetilde{X}^{(i-1)}Q_j^{(i)}\|_2 \le \delta\sqrt{2\pi pm \log N_{i-1}} \max_{1 \le j \le N_{i-1}} \|X_j^{(i-1)}\|_2$$
$$+ \left(N_{i-1}\|W^{(i)}\|_{\max}\|P^{(i-1)}\|_2^{r-1} + \delta\sqrt{2\pi pm \log N_{i-1}}\right) \max_{1 \le j \le N_{i-1}} \|X^{(i-2)}W_j^{(i-1)} - \widetilde{X}^{(i-2)}Q_j^{(i-1)}\|_2$$

holds with probability exceeding $1 - \frac{\sqrt{2}mN_i}{N_{i-1}^p}$. $\qquad\square$

Applying Theorem D.3 inductively for all layers, one can obtain an error bound for quantizing the whole neural network. Now we are ready to prove Theorem 3.1.

*Proof.* (a) For $1 \le j \le N_L$, by 7, we have

$$\Phi(X)_j = X_j^{(L)} = \rho(X^{(L-1)}W_j^{(L)}) \quad \text{and} \quad \widetilde{\Phi}(X)_j = \widetilde{X}_j^{(L)} = \rho(\widetilde{X}^{(L-1)}Q_j^{(L)})$$

where $W_j^{(L)}$ and $Q_j^{(L)}$ are the $j$-th neuron in the $L$-th layer and its quantized version respectively. It follows from part (a) of Theorem D.3 with $i = L$ that

$$\max_{1 \le j \le N_L} \|\Phi(X)_j - \widetilde{\Phi}(X)_j\|_2 = \max_{1 \le j \le N_L} \|\rho(X^{(L-1)}W_j^{(L)}) - \rho(\widetilde{X}^{(L-1)}Q_j^{(L)})\|_2$$
$$\le \max_{1 \le j \le N_L} \|X^{(L-1)}W_j^{(L)} - \widetilde{X}^{(L-1)}Q_j^{(L)}\|_2$$
$$\le \delta\sqrt{2\pi pm \log N_{L-1}} \max_{1 \le j \le N_{L-1}} \|X_j^{(L-1)}\|_2$$
$$+ \delta\sqrt{2\pi pm \log N_{L-1}} \max_{1 \le j \le N_{L-1}} \|X^{(L-2)}W_j^{(L-1)} - \widetilde{X}^{(L-2)}Q_j^{(L-1)}\|_2.$$

holds with probability at least $1 - \frac{\sqrt{2}mN_L}{N_{L-1}^p}$. Moreover, by applying part (a) of Theorem D.3 with $i = L - 1$ to the result above, we obtain that

$$\max_{1 \le j \le N_L} \|\Phi(X)_j - \widetilde{\Phi}(X)_j\|_2 \le \delta\sqrt{2\pi pm \log N_{L-1}} \max_{1 \le j \le N_{L-1}} \|X_j^{(L-1)}\|_2 + 2\pi pm\delta^2$$
$$\times \sqrt{\log N_{L-1} \log N_{L-2}} \left(\max_{1 \le j \le N_{L-2}} \|X_j^{(L-2)}\|_2 + \max_{1 \le j \le N_{i-1}} \|X^{(i-2)}W_j^{(i-1)} - \widetilde{X}^{(i-2)}Q_j^{(i-1)}\|_2\right)$$

holds with probability at least $1 - \frac{\sqrt{2}mN_L}{N_{L-1}^p} - \frac{\sqrt{2}mN_{L-1}}{N_{L-2}^p}$. Repeating this argument inductively for $i = L - 2, L - 3, \ldots, 1$, one can derive

$$\max_{1 \le j \le N_L} \|\Phi(X)_j - \widetilde{\Phi}(X)_j\|_2 \le \sum_{i=0}^{L-1} (2\pi pm\delta^2)^{\frac{L-i}{2}} \left(\prod_{k=i}^{L-1} \log N_k\right)^{\frac{1}{2}} \max_{1 \le j \le N_i} \|X_j^{(i)}\|_2$$

with probability at least $1 - \sum_{i=1}^{L} \frac{\sqrt{2}mN_i}{N_{i-1}^p}$.

(b) The proof of 23 is similar to the one we had in part (a) except that we need to use part (b) of Theorem D.3 this time. Indeed, for the case of $i = L$,

$$\max_{1 \le j \le N_L} \|\Phi(X)_j - \widetilde{\Phi}(X)_j\|_2 = \max_{1 \le j \le N_L} \|\rho(X^{(L-1)}W_j^{(L)}) - \rho(\widetilde{X}^{(L-1)}Q_j^{(L)})\|_2$$
$$\le \max_{1 \le j \le N_L} \|X^{(L-1)}W_j^{(L)} - \widetilde{X}^{(L-1)}Q_j^{(L)}\|_2$$
$$\le \delta\sqrt{2\pi pm \log N_{L-1}} \max_{1 \le j \le N_{L-1}} \|X_j^{(L-1)}\|_2 + \left(N_{L-1}\|W^{(L)}\|_{\max}\|P^{(L-1)}\|_2^{r-1}\right.$$
$$+ \left.\delta\sqrt{2\pi pm \log N_{L-1}}\right) \max_{1 \le j \le N_{L-1}} \|X^{(L-2)}W_j^{(L-1)} - \widetilde{X}^{(L-2)}Q_j^{(L-1)}\|_2$$

holds with probability exceeding $1 - \frac{\sqrt{2}mN_L}{N_{L-1}^p}$. Then 23 follows by inductively using part (b) of Theorem D.3 with $i = L - 1, L - 2, \ldots, 1$. $\qquad\square$

# E  PROOF OF COROLLARY 3.2

*Proof.* $(a)$ Conditioning on $X^{(i-1)}$, the function $f(z) := \|\rho(X^{(i-1)}z)\|_2$ is Lipschitz with Lipschitz constant $L_f := \|X^{(i-1)}\|_2 \leq \|X^{(i-1)}\|_F$ and $\|X_j^{(i)}\|_2 = \|\rho(X^{(i-1)}W_j^{(i)})\|_2 = f(W_j^{(i)})$ with $W_j^{(i)} \sim \mathcal{N}(0, I)$. Applying Lemma C.4 to $f$ with $X = W_j^{(i)}$, Lipschitz constant $L_f$, and $\alpha = \sqrt{2p\log N_{i-1}}\|X^{(i-1)}\|_F$, we obtain

$$P\Big(\big|\|X_j^{(i)}\|_2 - \mathbb{E}(\|X_j^{(i)}\|_2 \mid X^{(i-1)})\big| \leq \sqrt{2p\log N_{i-1}}\|X^{(i-1)}\|_F \;\Big|\; X^{(i-1)}\Big) \geq 1 - \frac{2}{N_{i-1}^p}. \quad (75)$$

Using Jensen's inequality and the identity $\mathbb{E}(\|\rho(X^{(i-1)}W_j^{(i)})\|_2^2 \mid X^{(i-1)}) = \frac{1}{2}\|X^{(i-1)}\|_F^2$, we have

$$\mathbb{E}(\|X_j^{(i)}\|_2 \mid X^{(i-1)}) \leq \Big(\mathbb{E}(\|X_j^{(i)}\|_2^2 \mid X^{(i-1)})\Big)^{\frac{1}{2}}$$
$$= \Big(\mathbb{E}(\|\rho(X^{(i-1)}W_j^{(i)})\|_2^2 \mid X^{(i-1)})\Big)^{\frac{1}{2}}$$
$$= \frac{1}{\sqrt{2}}\|X^{(i-1)}\|_F.$$

It follows from the inequality above and 75 that, conditioning on $X^{(i-1)}$,

$$\|X_j^{(i)}\|_2 \leq \Big(\frac{1}{\sqrt{2}} + \sqrt{2p\log N_{i-1}}\Big)\|X^{(i-1)}\|_F \leq 2\sqrt{p\log N_{i-1}}\|X^{(i-1)}\|_F$$

holds with probability at least $1 - \frac{2}{N_{i-1}^p}$. Conditioning on $X^{(i-1)}$ and taking a union bound over $1 \leq j \leq N_i$, with probability exceeding $1 - \frac{2N_i}{N_{i-1}^p}$, we have

$$\|X^{(i)}\|_F \leq \sqrt{N_i}\max_{1\leq j\leq N_i}\|X_j^{(i)}\|_2 \leq 2\sqrt{pN_i\log N_{i-1}}\|X^{(i-1)}\|_F. \quad (76)$$

Applying 76 for indices $i, i-1, \dots, 1$ recursively, we obtain 24.

$(b)$ Applying Jensen's inequality and Proposition C.5, we have

$$\mathbb{E}(\|X_j^{(i)}\|_2^2 \mid X^{(i-1)}) = \mathbb{E}(\|\rho(X^{(i-1)}W_j^{(i)})\|_2^2 \mid X^{(i-1)})$$
$$\geq \Big(\mathbb{E}(\|\rho(X^{(i-1)}W_j^{(i)})\|_2 \mid X^{(i-1)})\Big)^2$$
$$\geq \frac{\mathrm{tr}(X^{(i-1)}X^{(i-1)\top})}{2\pi}$$
$$= \frac{\|X^{(i-1)}\|_F^2}{2\pi}.$$

By the law of total expectation, we obtain $\mathbb{E}_\Phi\|X_j^{(i)}\|_2^2 \geq \frac{1}{2\pi}\mathbb{E}_\Phi\|X^{(i-1)}\|_F^2$ and thus

$$\mathbb{E}_\Phi\|X^{(i)}\|_F^2 = \sum_{j=1}^{N_i}\mathbb{E}_\Phi\|X_j^{(i)}\|_2^2 \geq \frac{N_i}{2\pi}\mathbb{E}_\Phi\|X^{(i-1)}\|_F^2. \quad (77)$$

Then 25 follows immediately by applying 77 recursively. $\qquad\square$

Now we are ready to evaluate the relative error in 26. It follows from 22 and the Cauchy-Schwarz inequality that, with high probability,

$$\frac{\|\Phi(X) - \widetilde{\Phi}(X)\|_F^2}{\mathbb{E}_\Phi\|\Phi(X)\|_F^2} \leq \frac{N_L \max_{1\leq j\leq N_L}\|\Phi(X)_j - \widetilde{\Phi}(X)_j\|_2^2}{\mathbb{E}_\Phi\|\Phi(X)\|_F^2}$$
$$\leq \frac{N_L}{\mathbb{E}_\Phi\|\Phi(X)\|_F^2}\Big(\sum_{i=0}^{L-1}(2\pi pm\delta^2)^{\frac{L-i}{2}}\Big(\prod_{k=i}^{L-1}\log N_k\Big)^{\frac{1}{2}}\max_{1\leq j\leq N_i}\|X_j^{(i)}\|_2\Big)^2$$
$$\leq \frac{LN_L}{\mathbb{E}_\Phi\|\Phi(X)\|_F^2}\sum_{i=0}^{L-1}(2\pi pm\delta^2)^{L-i}\Big(\prod_{k=i}^{L-1}\log N_k\Big)\max_{1\leq j\leq N_i}\|X_j^{(i)}\|_2^2. \quad (78)$$

By Corollary 3.2, $\max_{1 \leq j \leq N_i} \|X_j^{(i)}\|_2^2 \leq (4p)^i \|X\|_F^2 \log N_0 \prod_{k=1}^{i-1} (N_k \log N_k)$ with high probability, and $\mathbb{E}_\Phi \|\Phi(X)\|_F^2 = \mathbb{E}_\Phi \|X^{(L)}\|_F^2 \geq \frac{\|X\|_F^2}{(2\pi)^L} \prod_{k=1}^L N_k$. Plugging these results into 78,

$$
\begin{aligned}
\frac{\|\Phi(X) - \widetilde{\Phi}(X)\|_F^2}{\mathbb{E}_\Phi \|\Phi(X)\|_F^2} &\leq L(2\pi)^L \Big( \prod_{k=0}^L \log N_k \Big) \sum_{i=0}^{L-1} \frac{(2\pi p m \delta^2)^{L-i} (4p)^i}{\prod_{k=i}^{L-1} N_k} \\
&\lesssim \Big( \prod_{k=0}^L \log N_k \Big) \sum_{i=0}^{L-1} \prod_{k=i}^{L-1} \frac{m}{N_k}
\end{aligned}
\tag{79}
$$

gives an upper bound on the relative error of quantization method in Algorithm 1 with perfect data alignment. Further, if we assume $N_{\min} \leq N_i \leq N_{\max}$ for all $i$, and $2m \leq N_{\min}$, then 79 becomes

$$
\begin{aligned}
\frac{\|\Phi(X) - \widetilde{\Phi}(X)\|_F^2}{\mathbb{E}_\Phi \|\Phi(X)\|_F^2} &\lesssim (\log N_{\max})^{L+1} \sum_{i=0}^{L-1} \Big( \frac{m}{N_{\min}} \Big)^{L-i} \\
&\lesssim \frac{m(\log N_{\max})^{L+1}}{N_{\min}}.
\end{aligned}
$$

# F  PROOF OF THEOREM 4.1

*Proof.* Fix a neuron $w := W_j^{(i)} \in \mathbb{R}^{N_{i-1}}$ for some $1 \leq j \leq N_i$. By our assumption 27, the aligned weights $\widetilde{w}$ satisfy $\|\widetilde{w}\|_\infty \leq \frac{1}{2} K^{(i)} \delta$. Then, we perform the iteration 15 in Algorithm 1 with perfect data alignment. At the $t$-th step, similar to 64, 66, and 68, we have

$$
\widetilde{u}_t = P_{\widetilde{X}_t^{(i-1)\perp}}(h_t) + (v_t - \widetilde{q}_t) \widetilde{X}_t^{(i-1)}
$$

where

$$
h_t = \widetilde{u}_{t-1} + \widetilde{w}_t \widetilde{X}_t^{(i-1)}, \quad v_t = \frac{\langle \widetilde{X}_t^{(i-1)}, h_t \rangle}{\|\widetilde{X}_t^{(i-1)}\|_2^2}, \quad \text{and} \quad \widetilde{q}_t = \mathcal{Q}_{\mathrm{StocQ}}(v_t).
\tag{80}
$$

If $t = 1$, then $h_1 = \widetilde{w}_1 \widetilde{X}_1^{(i-1)}$, $v_1 = \widetilde{w}_1$, and $\widetilde{q}_1 = \mathcal{Q}_{\mathrm{StocQ}}(v_1)$. Since $|v_1| = |\widetilde{w}_1| \leq \|\widetilde{w}\|_\infty \leq \frac{1}{2} K^{(i)} \delta$, we get $|v_1 - \widetilde{q}_1| \leq \delta$ and the proof technique used for the case $t = 1$ in Lemma D.1 can be applied here to conclude that $\widetilde{u}_1 \leq_{\mathrm{cx}} \mathcal{N}(0, \sigma_1^2 I)$ with $\sigma_1^2 = \frac{\pi \delta^2}{2} \|\widetilde{X}_1^{(i-1)}\|_2^2$. Next, for $t \geq 2$, assume that $\widetilde{u}_{t-1} \leq_{\mathrm{cx}} \mathcal{N}(0, \sigma_{t-1}^2 I)$ holds where $\sigma_{t-1}^2 = \frac{\pi \delta^2}{2} \max_{1 \leq j \leq t-1} \|\widetilde{X}_j^{(i-1)}\|_2^2$ is defined as in Lemma D.1. It follows from 80 and Lemma A.4 that

$$
|v_t| = \Big| \frac{\langle \widetilde{X}_t^{(i-1)}, \widetilde{u}_{t-1} \rangle}{\|\widetilde{X}_t^{(i-1)}\|_2^2} + \widetilde{w}_t \Big| \leq \Big| \frac{\langle \widetilde{X}_t^{(i-1)}, \widetilde{u}_{t-1} \rangle}{\|\widetilde{X}_t^{(i-1)}\|_2^2} \Big| + \|\widetilde{w}\|_\infty \leq \Big| \frac{\langle \widetilde{X}_t^{(i-1)}, \widetilde{u}_{t-1} \rangle}{\|\widetilde{X}_t^{(i-1)}\|_2^2} \Big| + \frac{1}{2} K^{(i)} \delta
$$

with $\frac{\langle \widetilde{X}_t^{(i-1)}, \widetilde{u}_{t-1} \rangle}{\|\widetilde{X}_t^{(i-1)}\|_2^2} \leq_{\mathrm{cx}} \mathcal{N} \Big( 0, \frac{\sigma_{t-1}^2}{\|\widetilde{X}_t^{(i-1)}\|_2^2} \Big)$. Then we have, by Lemma C.2, that

$$
\mathrm{P}(|v_t| \leq K^{(i)} \delta) \geq \mathrm{P} \Big( \Big| \frac{\langle \widetilde{X}_t^{(i-1)}, \widetilde{u}_{t-1} \rangle}{\|\widetilde{X}_t^{(i-1)}\|_2^2} \Big| \leq \frac{1}{2} K^{(i)} \delta \Big) \geq 1 - \sqrt{2} \exp \Big( -\frac{(K^{(i)} \delta)^2}{16 \sigma_{t-1}^2} \|\widetilde{X}_t^{(i-1)}\|_2^2 \Big).
$$

On the event $\{|v_t| \leq K^{(i)} \delta\}$, we can quantize $v_t$ as if the quantizer $\mathcal{Q}_{\mathrm{StocQ}}$ used the infinite alphabet $\mathcal{A}_\infty^\delta$. So $\widetilde{u}_t \leq_{\mathrm{cx}} \mathcal{N}(0, \sigma_t^2 I)$. Therefore, applying a union bound,

$$
\mathrm{P} \Big( \widetilde{u}_{N_{i-1}} \leq_{\mathrm{cx}} \mathcal{N}(0, \sigma_{N_{i-1}}^2 I) \Big) \geq 1 - \sqrt{2} \sum_{t=2}^{N_{i-1}} \exp \Big( -\frac{(K^{(i)} \delta)^2}{16 \sigma_{t-1}^2} \|\widetilde{X}_t^{(i-1)}\|_2^2 \Big).
\tag{81}
$$

Conditioning on the event above, that $\widetilde{u}_{N_{i-1}} \leq_{\mathrm{cx}} \mathcal{N}(0, \sigma_{N_{i-1}}^2 I)$, Lemma C.2 yields for $\gamma \in (0, 1]$

$$
\mathrm{P} \Big( \|\widetilde{u}_{N_{i-1}}\|_\infty \leq 2 \sigma_{N_{i-1}} \sqrt{\log(\sqrt{2} m / \gamma)} \Big) \geq 1 - \gamma.
$$

Setting $\gamma = \sqrt{2} m N_{i-1}^{-p}$ and recalling 19, we obtain that

$$
\|X^{(i-1)} W_j^{(i)} - \widetilde{X}^{(i-1)} Q_j^{(i)}\|_2 = \|\widetilde{u}_{N_{i-1}}\|_2 \leq \sqrt{m} \|\widetilde{u}_{N_{i-1}}\|_\infty \leq 2 \sigma_{N_{i-1}} \sqrt{m p \log N_{i-1}}
\tag{82}
$$

holds with probability at least $1 - \frac{\sqrt{2}m}{N_{i-1}^p}$. Combining 81 and 82, for each $1 \leq j \leq N_i$,

$$\|X^{(i-1)}W_j^{(i)} - \widetilde{X}^{(i-1)}Q_j^{(i)}\|_2 \leq 2\sigma_{N_{i-1}}\sqrt{mp\log N_{i-1}} = \delta\sqrt{2\pi pm\log N_{i-1}} \max_{1 \leq j \leq N_{i-1}} \|\widetilde{X}_j^{(i-1)}\|_2$$

holds with probability exceeding $1 - \frac{\sqrt{2}m}{N_{i-1}^p} - \sqrt{2}\sum_{t=2}^{N_{i-1}} \exp\left(-\frac{(K^{(i)}\delta)^2}{16\sigma_{t-1}^2}\|\widetilde{X}_t^{(i-1)}\|_2^2\right)$. Taking a union bound over all $1 \leq j \leq N_i$, we have

$$\mathrm{P}\left(\max_{1 \leq j \leq N_i} \|X^{(i-1)}W_j^{(i)} - \widetilde{X}^{(i-1)}Q_j^{(i)}\|_2 \leq \delta\sqrt{2\pi pm\log N_{i-1}} \max_{1 \leq j \leq N_{i-1}} \|\widetilde{X}_j^{(i-1)}\|_2\right)$$

$$\geq 1 - \frac{\sqrt{2}mN_i}{N_{i-1}^p} - \sqrt{2}N_i \sum_{t=2}^{N_{i-1}} \exp\left(-\frac{(K^{(i)}\delta)^2}{16\sigma_{t-1}^2}\|\widetilde{X}_t^{(i-1)}\|_2^2\right)$$

$$\geq 1 - \frac{\sqrt{2}mN_i}{N_{i-1}^p} - \sqrt{2}N_i \sum_{t=2}^{N_{i-1}} \exp\left(-\frac{(K^{(i)})^2\|\widetilde{X}_t^{(i-1)}\|_2^2}{8\pi\max_{1 \leq j \leq t-1}\|\widetilde{X}_j^{(i-1)}\|_2^2}\right).$$

$\square$

## G    PROOF OF THEOREM 4.2

*Proof.* Pick a neuron $w := W_j^{(i)} \in \mathbb{R}^{N_{i-1}}$ for some $1 \leq j \leq N_i$. Then we have $w \sim \mathcal{N}(0, I)$ and since we are using Algorithm 1 with perfect data alignment, we must work with the resulting $\widetilde{w}$, the solution of 18. Applying Proposition C.11 to $w$ with $X = X^{(i-1)}$ and $\widetilde{X} = \widetilde{X}^{(i-1)}$, we obtain

$$\mathrm{P}\left(\|\widetilde{w}\|_\infty \leq \eta^{(i-1)}\sqrt{2p\log N_{i-1}}\right) \geq 1 - \frac{2}{N_{i-1}^{p-1}},$$

so that using 30 gives

$$\mathrm{P}\left(\|\widetilde{w}\|_\infty \leq \frac{1}{2}K^{(i)}\delta\right) \geq 1 - \frac{2}{N_{i-1}^{p-1}}. \tag{83}$$

Conditioning on the event $\{\|\widetilde{w}\|_\infty \leq \frac{1}{2}K^{(i)}\delta\}$ and applying exactly the same argument in Theorem 4.1,

$$\|X^{(i-1)}W_j^{(i)} - \widetilde{X}^{(i-1)}Q_j^{(i)}\|_2 \leq \delta\sqrt{2\pi pm\log N_{i-1}} \max_{1 \leq j \leq N_{i-1}} \|\widetilde{X}_j^{(i-1)}\|_2 \tag{84}$$

holds with probability exceeding $1 - \frac{\sqrt{2}m}{N_{i-1}^p} - \sqrt{2}\sum_{t=2}^{N_{i-1}} \exp\left(-\frac{(K^{(i)})^2\|\widetilde{X}_t^{(i-1)}\|_2^2}{8\pi\max_{1 \leq j \leq t-1}\|\widetilde{X}_j^{(i-1)}\|_2^2}\right)$. Combining 83 and 84, and taking a union bound over all $1 \leq j \leq N_i$, we obtain 31. $\square$

Table 1: Top-1/Top-5 validation accuracy for SPFQ on ImageNet.

| Model | $m$ | $b$ | $C$ | Quant Acc (%) | Ref Acc (%) | Acc Drop (%) |
|-------|-----|-----|-----|---------------|-------------|--------------|
| VGG-16 | 1024 | 4 | 1.02 | 70.48/89.77 | 71.59/90.38 | 1.11/0.61 |
| | | 5 | 1.23 | 71.08/90.15 | 71.59/90.38 | 0.51/0.23 |
| | | 6 | 1.26 | 71.24/90.37 | 71.59/90.38 | 0.35/0.01 |
| ResNet-18 | 2048 | 4 | 0.91 | 67.36/87.74 | 69.76/89.08 | 2.40/1.34 |
| | | 5 | 1.32 | 68.79/88.77 | 69.76/89.08 | 0.97/0.31 |
| | | 6 | 1.68 | 69.43/88.96 | 69.76/89.08 | 0.33/0.12 |
| ResNet-50 | 2048 | 4 | 1.10 | 73.37/91.61 | 76.13/92.86 | 2.76/1.25 |
| | | 5 | 1.62 | 75.05/92.43 | 76.13/92.86 | 1.08/0.43 |
| | | 6 | 1.98 | 75.66/92.67 | 76.13/92.86 | 0.47/0.19 |

# H    EXPERIMENTS

In this section, we test the performance of SPFQ on the ImageNet classification task and compare it with the non-random scheme GPFQ in (Zhang et al., 2023). In particular, we adopt the version of SPFQ corresponding to 11, i.e., Algorithm 1 using approximate data alignment with order $r = 1$. Note that the GPFQ algorithm runs the same iterations as in 11 except that $\mathcal{Q}_{\text{StocQ}}$ is substituted with a non-random quantizer $\mathcal{Q}_{\text{DetQ}}$, so the associated iterations are given by

$$
\begin{cases}
u_0 = 0 \in \mathbb{R}^m, \\
q_t = \mathcal{Q}_{\text{DetQ}}\left( \frac{\langle \widetilde{X}_t^{(i-1)}, u_{t-1} + w_t X_t^{(i-1)} \rangle}{\|\widetilde{X}_t^{(i-1)}\|_2^2} \right), \\
u_t = u_{t-1} + w_t X_t^{(i-1)} - q_t \widetilde{X}_t^{(i-1)}
\end{cases}
\tag{85}
$$

where $\mathcal{Q}_{\text{DetQ}}(z) := \arg\min_{p \in \mathcal{A}} |z - p|$. For ImageNet data, we consider ILSVRC-2012 (Deng et al., 2009), a 1000-category dataset with over 1.2 million training images and 50 thousand validation images. Additionally, we resize all images to $256 \times 256$ and use the normalized $224 \times 224$ center crop, which is a standard procedure. The evaluation metrics we choose are top-1 and top-5 accuracy of the quantized models on the validation dataset. As for the neural network architectures, we quantize all layers of VGG-16 (Simonyan & Zisserman, 2015), ResNet-18 and ResNet-50 (He et al., 2016), which are pretrained 32-bit floating point neural networks provided by torchvision in PyTorch (Paszke et al., 2019). Moreover, we fuse the batch normalization (BN) layer with the convolutional layer, and freeze the BN statistics before quantization.

Since the major difference between SPFQ in 11 and GPFQ in 85 is the choice of quantizers, we will follow the experimental setting for alphabets used in (Zhang et al., 2023). Specifically, we use batch size $m$, fixed bits $b \in \mathbb{N}$ for all the layers, and quantize each $W^{(i)} \in \mathbb{R}^{N_{i-1} \times N_i}$ with midtread alphabets $\mathcal{A} = \mathcal{A}_K^\delta$ as in 5, where level $K$ and step size $\delta$ are given by

$$
K = 2^{b-1}, \quad \delta = \delta^{(i)} := \frac{C}{2^{b-1} N_i} \sum_{1 \le j \le N_i} \|W_j^{(i)}\|_\infty.
\tag{86}
$$

Here, $C > 0$ is a constant that is only dependent on bitwidth $b$, determined by grid search with cross-validation, and fixed across layers, and across batch-sizes. One can, of course, expect to do better by using different values of $C$ for different layers but we refrain from doing so, as our main goal here is to demonstrate the performance of SPFQ even with minimal fine-tuning.

In Table 1, for different combinations of $m$, $b$, and $C$, we present the corresponding top-1/top-5 validation accuracy of quantized networks using SPFQ in the first column, while the second and thrid columns give the validation accuracy of unquantized models and the accuracy drop due to quantization respectively. We observe that, for all three models, the quantization accuracy is improved as the number of bits $b$ increases, and SPFQ achieves less than $0.5\%$ top-1 accuracy loss while using 6 bits.

Next, in Figure 1, we compare SPFQ against GPFQ by quantizing the three models in Table 1. These figures illustrate that GPFQ has better performance than that of SPFQ when $b = 3, 4$ and $m$ is small. This is not particularly surprising, as $\mathcal{Q}_{\text{DetQ}}$ deterministically rounds its argument to the nearest alphabet element instead of performing a random rounding like $\mathcal{Q}_{\text{StocQ}}$. However, as the batch size $m$ increases, the accuracy gap between GPFQ and SPFQ diminishes. Indeed, for VGG-16 and ResNet-18, SPFQ outperforms GPFQ when $b = 6$. Further, we note that, for both SPFQ and GPFQ, one can obtain higher quantization accuracy by taking larger $m$ but the extra improvement that results from increasing the batch size rapidly decreases.

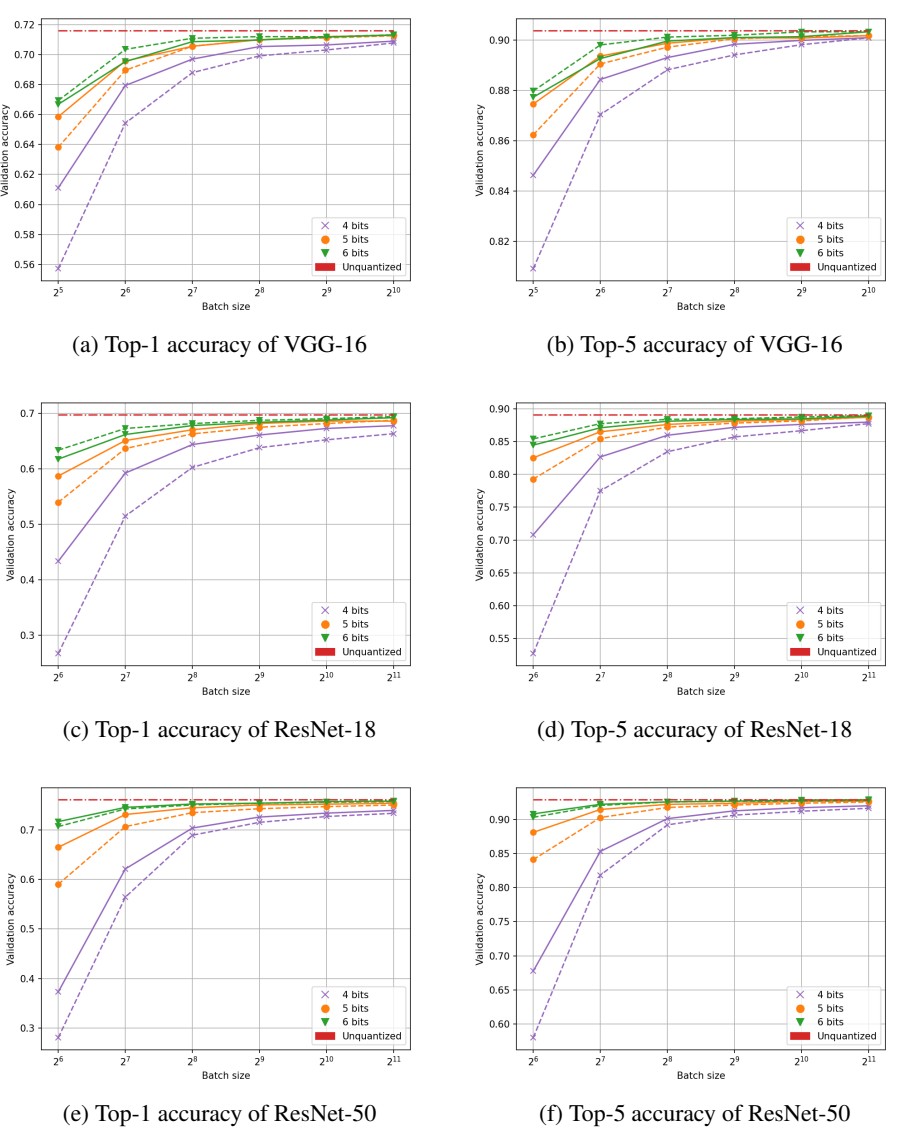

(a) Top-1 accuracy of VGG-16

(b) Top-5 accuracy of VGG-16

(c) Top-1 accuracy of ResNet-18

(d) Top-5 accuracy of ResNet-18

(e) Top-1 accuracy of ResNet-50

(f) Top-5 accuracy of ResNet-50

Figure 1: Top-1 and Top-5 validation accuracy for SPFQ (dashed lines) and GPFQ (solid lines) on ImageNet.

