# OpenReview forum: "SPFQ: A Stochastic Algorithm and Its Error Analysis for Neural Network Quantization"
_ICLR.cc/2024/Conference — Submitted to ICLR 2024_

### Official Review · Reviewer_dz6f · 2023-10-30

**Soundness:** 2 fair
**Presentation:** 2 fair
**Contribution:** 2 fair
**Rating:** 3
**Confidence:** 3

**Summary:**

The paper presents a fast stochastic approach for quantizing the weights of neural networks with full-network error bounds.

**Strengths:**

The authors for the first time propose error bounds for quantizing an entire L-layer neural network under minimal assumptions on weights and input data in the infinite alphabet case.

**Weaknesses:**

If I understand correctly, in Figure 1 and Table 1, the authors quantize only the weights of neural networks, which cannot bring out the benefits of quantization. What this means is that to fully exploit quantization for convolutional networks, it is necessary to quantize both weights and activations of neural networks because running convolutional networks is compute-bound.


Furthermore, more experiments on lightweight models such as MobileNetV2 are required due to the fact that it is well known that performance degradation is severe when quantizing MobileNetV2. To validate whether SPFQ is really powerful or not, the experiment on MobileNetV2 is necessary.

**Questions:**

N/A

---

> ### Author Response · Authors · 2023-11-21
>
> **If I understand correctly, in Figure 1 and Table 1, the authors quantize only the weights of neural networks, which cannot bring out the benefits of quantization. What this means is that to fully exploit quantization for convolutional networks, it is necessary to quantize both weights and activations of neural networks because running convolutional networks is compute-bound.**
>
> We thank the reviewer for the opportunity to clarify this misconception. Quantizaiton of activations is mathematically rather simple, while the quantization of weights is difficult. To illustrate, consider that our error bounds hold for any non-linearity that is a Lipschitz function. Now consider, a (for example) single neuron in the neural network where the activations are quantized. It would be modeled as $Q(\rho(Xw))$, where $Q$ is the scalar quantizer that maps the activations to their quantized values. Since the composition $\tilde{\rho}:=Q\circ\rho$ is a Lipschitz function (whenever $\rho$ is), our results hold verbatim with all activations quantized! If the activations are finely-quantized, then the resulting error bound does not change much. Indeed, this is in-line with what is typically done in practice, where a greater number of bits is used to quantize activations compared to weights. We explain this at the top of page 3 (highlighted in blue).
>
> **Furthermore, more experiments on lightweight models such as MobileNetV2 are required due to the fact that it is well known that performance degradation is severe when quantizing MobileNetV2. To validate whether SPFQ is really powerful or not, the experiment on MobileNetV2 is necessary.**
>
> As previously noted, we view this paper as a theory focused paper. However, we point out that MobileNetV2 is a highly compressed neural network, meaning that it has very little redundancy in its architecture. As such it is natural that quantizing it further would lead to performance degradation.

---

> > ### Comment · Reviewer_dz6f · 2023-11-22
> >
> > Thank you for the response.
> >
> > However, I cannot figure out why the quantization of activations is simple. For example, when empirically comparing quantizing only weights to $4$-bit with quantizing only activations to $4$-bit, the performance of the former would be much better than that of the latter.
> >
> > Although MobileNetV2 is a lightweight model, it is also important to quantize such a lightweight model well in the field of quantization because such a lightweight model can be deployed to resource-limited devices more efficiently via quantization.
> >
> > For these reasons, I keep my score.

---

> > > ### Author Response · Authors · 2023-11-22
> > >
> > > We said it was mathematically rather simple to quantize activations, and not that it had a more favorable or less favorable rate-distortion relationship compared to weight quantization. When quantizing a neural network, you can only use sophisticated algorithms to quantize the weights since you do it once and fix the weights forever. Moreover, you can achieve significant gains by doing so. With activations, on the other hand, you have to quantize them every time a new input is fed through the network, and you don’t have any control over the input. That means that, typically, the best you can do is round the activations. It is also essentially the fastest thing you can do, from a computational perspective. To reiterate, with weights, rounding the weights is not the best you can do, with activations there is not much more you can do.
> > >
> > > The mathematical content of our statement was that if you quantize the activations by rounding them, our theory still holds with barely any changes, and that was in response to your comment about activation quantization not being covered by our theory.
> > >
> > > Regarding MobileNetV2, we don’t disagree about it, or any other network becoming more lightweight after quantization.

---

### Official Review · Reviewer_xzB6 · 2023-10-31

**Soundness:** 3 good
**Presentation:** 3 good
**Contribution:** 3 good
**Rating:** 6
**Confidence:** 3

**Summary:**

The paper studies model compression through weight quantization. It introduces a new stochastic quantization framework, called (SPFQ), by employing randomness in the quantizer. SPFQ is consisted of a data-alignment phase and a quantization phase. SPFQ with approximate data alignment has a computational complexity that scales linearly in the number of parameters of the neural network. The paper also obtained a first error bounds for quantizing an L-layer neural network, under an infinite alphabet condition and limited assumptions on the weights and input data. Through experiments, the authors applied their quantization to the weights of several neural network architectures that are trained for classification tasks on the ImageNet dataset. The simulations indicated a small loss of accuracy compared to unquantized models.

**Strengths:**

-stochastic quantization appears to be a novel idea that has not been tried for the model compression although It was used in stochastic gradient(or weight parameter) quantization for communication in both distributed training and federated learning.

-the analysis of quantization error appears to be rigorous and correct, despite having some limiting but mild assumptions.

-it appears that the analysis is applicable with any non-linearities in activation function as long as it is a Lipschitz function, after properly changing the values of constants.

-it’s data alignment phase has a linear complexity which is improvement on deterministic quantization algorithms that require solving optimization problems, and hence usually having higher order complexity in the number of parameters.

**Weaknesses:**

-the error analysis was not validated by the simulations. Simulations only reflect the accuracy of the quantizer compare to unquantized method but it does not say anything about how good the error analysis is.

-experiments does not use any baseline method for comparison with the proposed method. Only unquantized model accuracy was compared. This is unacceptable.

**Questions:**

-please provide simulation results confirming how tight the error bounds.
-please provide discussion as to how the error analysis can help with the design of better quantizer. Is there a lesson to be learned from error bound that one can leverage to improve the quantizer design.
-please provide simulation results on imagenet dataset comparing the performance of the proposed method with the state of art model quantizers.

-the error analysis implies that the error upper bound increases with batch size. However, simulations suggest otherwise. Can you explain the contradiction?

-would fine tuning improve the accuracy. Perhaps simulation results with fine tuning would be helpful too.

---

> ### Author Response · Authors · 2023-11-21
>
> We are glad to see that the review thinks the theoretical contributions of the paper are novel and important to the community.
>
> 1. **response to weaknesses:** Thank you for this comment, with which we strongly disagree. We believe that theoretical work is of value in machine learning. As such, we focused our attention in this paper on proving that a variant of GPFQ, namely SPFQ, can provide full-network error bounds. We provided a short numerical experiments section to validate that the method is viable in practice, and performs comparably to GPFQ which was extensively compared to other methods in the original GPFQ paper (J. Zhang, Y. Zhou, R. Saab, "Post-training quantization for neural networks with provable guarantees", SIAM Journal on Mathematics of Data Science, 2023). Indeed, the numerics suggest that SPFQ can outperform GPFQ in some cases. Due to its stochasticity, SPFQ if run multiple times, can potentially even provide significantly improved performance at the cost of multiple runs. While we appreciate the practical importance of such explorations, we note that in order to explore these issues one would need to run extensive numerical experiments (as done in the GPFQ paper). Given the purpose of this paper, and the computational resources needed for running such experiments, we choose to keep the focus on theory.
>
> 2. **response to questions:** This is a good question. We think that the phenomenon the reviewer noted is due to the fact that the test error is a sum of two terms (in the spirit of generalization error bounds). One term is due to the training error (which we control in this paper) and the other is due to the complexity of the hypothesis class. The first term typically increases with training data-size while the second decreases. We believe this explains the seeming contradiction.

---

### Official Review · Reviewer_nVek · 2023-11-01

**Soundness:** 3 good
**Presentation:** 2 fair
**Contribution:** 2 fair
**Rating:** 3
**Confidence:** 3

**Summary:**

The authors propose stochastic path-following quantization (SPFQ), a greedy algorithm that sequentially quantizes the rows of a neural network's weight matrices starting from the input layer. SPFQ quantizes weights "keeping previously quantized weights in mind": it keeps track of the quantization error it introduces after each step, and to quantize the next row of weights, it selects a weight setting that minimizes the total network quantization error. This way, at each step, it corrects for a "bad" setting for the quantized weights.

The basic version of SPFQ greedily minimizes the 2-norm of the quantization error induced for the pre-activations of the current hidden layer. The authors show their algorithm admits a decomposition into two stages: the first step finds unquantized weight settings that would correct for the total quantization error as best as possible, and the second step quantizes these weight settings.

Based on this decomposition, the authors propose two variants of SPFQ. The first variant completely eliminates the total quantization in the first step. Hence, the only remaining error arises from quantizing the new weight settings; however, it is more involved computationally. Thus, the second proposed variant is an iterative scheme that eliminates the total quantization error only approximately but is cheaper computationally.

Finally, the authors derive full-network error bounds for these two SPFQ variants.

**Strengths:**

SPFQ is a simple and elegant algorithm, and its motivation in section 2.2 and decomposition in section 2.3 are particularly nice. The authors' theoretical results are rigorously stated, and having checked appendices A and B in detail and skimmed over C-G, I am reasonably confident that their proofs are correct.

I also found both the main text and the appendix to be well-written and reasonably easy to follow.

**Weaknesses:**

My biggest concern regarding this paper is that, in its current form, it is a poor fit for ICLR as the authors do not showcase how their contributions are relevant to the ICLR community in any way. The paper would be a better fit for a more specialized venue, e.g., COLT or a statistics conference/journal.

I hold this opinion because, under the current narrative, the purpose of SPFQ is to allow the authors to derive the bounds in sections 3 and 4 and not to be a practical scheme (though I should emphasize that this criticism is in no way meant to imply that I don't find the bounds interesting). This issue appears at multiple levels in the paper:
- The original version of SPFQ is motivated by minimizing the 2-norm of the pre-activation's quantization error, from which authors then deviate with their two proposed variants (though the second variant contains the original as a special case). Since the variants are technically no longer controlling the 2-norm of the quantization error, and the authors do not comment on the benefits of deviating from the original formulation, this appears to indicate that their only purpose is that the error bounds in sections 3 and 4 can be derived for them.
- The authors do not numerically evaluate any of their bounds. Hence, also in part due to their complexity, it is impossible to gauge the bounds' tightness in any scenario.
- The usefulness of the relative error bound in Corollary 3.2 and related results seems very limited, and I am not convinced that they bear out their feature as one of the premier contributions in Eq (2). This is because the authors need to assume that the network weights follow a standard Gaussian distribution for their results to hold. This assumption usually holds before training (as a standard, but at least isotropic Gaussian is a commonly used prior for neural network weights) and certainly does not hold after. Thus, the authors' results regarding the number of bits required to quantize a network seem to only hold for untrained networks in practice.
- The authors do not state what SPFQ's intended purpose is: is it to compress a network or to speed up inference? This question is important because it determines the relevant benchmarks the method should be compared with.
- The experiments are relegated to Appendix H and are limited to evaluating SPFQ on a few small architectures and comparing it with only one other related work. In particular, there are no comparisons with state-of-the-art post-training quantization methods.
- Moreover, there are no ablation studies between the different SPFQ variants.

Writing-wise, while the paper is well-written, it is incomplete, as it is missing a conclusion/discussion section.

Finally, a related work I think the authors should mention is that of [2], which also explores the idea of sequentially correcting for poor stochastic quantization in a greedy/myopic way, though in the context of variational autoencoders.

## References
 - [1] Eric Lybrand and Rayan Saab. A greedy algorithm for quantizing neural networks. J. Mach. Learn. Res., 22:156–1, 2021
 - [2] Gergely Flamich, Marton Havasi and José Miguel Hernández Lobato. Compressing Images by Encoding Their Latent Representations with Relative Entropy Coding. In Advances of Neural Information Processing Systems, 2020.

**Questions:**

n/a

---

> ### Author Response · Authors · 2023-11-21
>
> 1. **"My biggest concern regarding this paper is that, in its current form, it is a poor fit for ICLR as the authors do not showcase how their contributions are relevant to the ICLR community in any way. The paper would be a better fit for a more specialized venue, e.g., COLT or a statistics conference/journal."**
>
> We  disagree with this comment, and find it somewhat troubling. Indeed, we believe, and we hope the reviewer agrees, that theoretical work is of value in machine learning. From the "non-exhaustive" list of ICLR's subject areas (taken from the Call for Papers), we find that "theoretical issues in deep learning" is clearly listed. So is "implementation issues, parallelization, software platforms, hardware". Our paper straddles this interface.
>
> 2. **"The original version of SPFQ is motivated by minimizing the 2-norm of the pre-activation's quantization error, ... this appears to indicate that their only purpose is that the error bounds in sections 3 and 4 can be derived for them."**
>
> Thank you for the opportunity to clarify. The data alignment is indeed for technical reasons related to the proof. That said, the reviewer is not correct that the variants are no longer controlling the 2-norm of the quantization error. They, in fact, are. This is clearly highlighted in Theorems 3.1, and 4.1 which control precisely the 2-norm of the quantization error.
>
> 3. **"The authors do not numerically evaluate any of their bounds. Hence, also in part due to their complexity, it is impossible to gauge the bounds' tightness in any scenario."**
>
> We derive full-neural network error bounds. To validate their tightness, we would need to run experiments with multiple neural networks of varying width, depth, and data-sizes. In other words, we would need to vary $m, N_i, i=1,...,L$ which would be very computationally expensive for real-world networks.
>
> 4. **The usefulness of the relative error bound in Corollary 3.2 and related results seems very limited, and I am not convinced that they bear out their feature as one of the premier contributions in Eq (2). This is because the authors need to assume that the network weights follow a standard Gaussian distribution for their results to hold. This assumption usually holds before training (as a standard, but at least isotropic Gaussian is a commonly used prior for neural network weights) and certainly does not hold after. Thus, the authors' results regarding the number of bits required to quantize a network seem to only hold for untrained networks in practice.**
>
> The purpose of Corollary 3.2 is just to illustrate the use of Theorems 3.1 and 4.1 (which provide absolute error bounds) to obtain relative error bounds when extra assumptions on the weights are available. In the corollary, we made the Gaussian weight assumption to illustrate this use. That said, there are regimes, e.g. lazy training (see "On lazy training in differentiable programming" by L. Chizat et al. and "Disentangling feature and lazy training in deep neural networks" by M. Geiger et al.), where the final values of the trained weights are very close to their initialization, which is itself Gaussian. This is very close to the setting of our corollary.
>
> 5. **The authors do not state what SPFQ's intended purpose is: is it to compress a network or to speed up inference? This question is important because it determines the relevant benchmarks the method should be compared with.**
>
> The purpose is to quantize the network so it can be implemented in hardware, more cheaply, and with faster inference times. Compression is an added benefit.
>
> 6. **The experiments are relegated to Appendix H and are limited to evaluating SPFQ on a few small architectures and comparing it with only one other related work. In particular, there are no comparisons with state-of-the-art post-training quantization methods. Moreover, there are no ablation studies between the different SPFQ variants.**
>
> We again highlight the importance of theoretical work in machine learning. We also point out that we compared SPFQ to GPFQ and found them quite similar for reasonable choices of $m$. In turn, the GPFQ paper (J. Zhang, Y. Zhou, R. Saab, "Post-training quantization for neural networks with provable guarantees", SIAM Journal on Mathematics of Data Science, 2023). has extensive comparisons to state of the art methods.
>
> 7. **Finally, a related work I think the authors should mention is that of [2], which also explores the idea of sequentially correcting for poor stochastic quantization in a greedy/myopic way, though in the context of variational autoencoders. [2] Gergely Flamich, Marton Havasi and José Miguel Hernández Lobato. Compressing Images by Encoding Their Latent Representations with Relative Entropy Coding. In Advances of Neural Information Processing Systems, 2020.**
>
> We thank the reviewer for the reference, but unfortunately could not see the connection.

---

> > ### Comment · Reviewer_nVek · 2023-11-21
> > **Response to the authors**
> >
> > I thank the authors for their elaborate response. I would like to make it clear that I find the authors' contributions interesting. However, after reading the authors' rebuttal, I am now more convinced that their paper is not a good fit for ICLR.
> >
> > I want to emphasize that I realize that one needs to think twice before claiming that a paper is not a good fit. I also do theoretical work, and thus, I would like to reassure the authors that my issue is certainly not that their contributions are theoretical. Instead, as I write in the first sentence of the weaknesses section of my review, I do not see how the paper is relevant to the ICLR community in its current form. I have no doubt the authors trust in the relevance and potential of their approach. However, I don't think the authors make a convincing case for anyone from the community to build on their work. This is because:
> >  - as is clear from the rebuttal, the authors do not intend to evaluate their bounds numerically (doing which could single-handedly make the paper relevant)
> >  - furthermore, they also do not demonstrate their bounds are non-vacuous in any setting. For example, assuming the range of the activations is bounded, the quantization error is trivially bounded by the square of the range's size. Is there any way to see that the authors' bounds outperform this baseline? Or are there other sensible baselines where the authors can demonstrate that their bounds are non-vacuous?
> >
> > Regarding point 2) of the rebuttal, the authors are right; the way I phrased my point did not have the intended meaning, and they have correctly responded to its actual meaning. I actually meant to write that in the original motivation of SPFQ, the quantity $c^*$ defined between Eqs (9) and (10) is the optimal choice to minimize the 2-norm of the error by definition. Hence, quantizing anything else (as done in the variants) seems suboptimal. This is the sense in which I meant that using the seemingly suboptimal variants only appears to have the purpose of allowing the authors to derive the respective bounds for them.

---

> > > ### Author Response · Authors · 2023-11-22
> > >
> > > We sincerely thank the reviewer for engaging with us, but it seems we will have to respectfully disagree on all counts. First, it is not that we do not wish to run numerical experiments for arbitrary reasons. Rather, we believe that additional numerical experiments will add little value to the paper above the experiments that we have already conducted. The point we’ve been trying to make is that the performance of SPFQ is quite similar to that of GPFQ (albeit with full-network error bounds), and GPFQ has been extensively numerically tested. Our decision is simply based on that, and the fact that our paper is already quite lengthy, and on the amount of time and computational resources it would take to conduct these experiments, which would certainly put us beyond the rebuttal period of ICLR.
> > >
> > > As to your second point, we thank you again for the opportunity to clarify misconceptions. Regarding showing that the bounds are not vacuous in any setting, we are surprised by your comment, as that was one of the points of doing the example with gaussian weights!  Consider for example equation (26) which shows a relative square error decay of $\approx \frac{m \cdot\mathrm{poly}(\log(N))}{N}$, where N is the width of the network. Now, consider even a single layer, and suppose that instead of quantizing the weights the way that we do, one simply rounds each weight to its nearest element in the alphabet. This would mean that  $$||Xw-Xq||^2_2 \leq (\sum_{i=1}^N ||X_i||_2  |w_i-q_i|)^2$$ is only bounded by $\delta^2 N^2 \max ||X_i||^2$ where $\delta$ is the quantization stepsize. The resulting relative error bound would not decay with N. In other words it would not take advantage of the overparametrization in the layer. On the other hand our bound does decay with N. In more detail, Lemma D.1 shows a logarithmic dependence of the square error on N, while a vacuous/trivial bound would depend linearly on N. Note that the lemma naturally incorporates the boundedness of the activations.

---

### Official Review · Reviewer_sBRp · 2023-11-04

**Soundness:** 3 good
**Presentation:** 3 good
**Contribution:** 2 fair
**Rating:** 5
**Confidence:** 4

**Summary:**

This work proposes a quantization algorithm for compressing pre-trained neural networks. The proposed scheme is a greedy algorithm that   quantizes weights sequentially. Given a set of quantized weights, the neural network activations are then found with those quantized weights, which is then weighted / aligned with the unquantized output via scaling. The effect of this scaling is incorporated in the weight, which gives the effective parameter value to be quantized. Since this is sequential, and at every step the algorithm tries to minimize the $\ell_2$-error between the outputs of the quantized weights and the original weights. this algorithm is referred to as "greedy".

Moreover, the quantizer used in this paper is a stochastic quantizer, wherein the quantization levels are decided, and the output of the quantizer is one of the two quantization points closest to the scalar input. The output is probabilisitic with a probability that is inversely proportional to the distance of the input to these points. Such a quantizer ensures that the output is unbiased, which ensures analytical tractability. Furthermore, when the quantizer is unsaturated, i.e., when the number of quantization levels in infinite, or the quantizer input lies in a pre-defined dynamic range, the variance the the quantization error is also bounded.

The authors use these two ideas in this work to derive error bounds for a multi-layer (deep) feed-forward neural neural network.

**Strengths:**

Quantization of neural networks is an important problem, provided the significant research impetus that is going into this domain in order to  make modern large neural networks (which can easily consist of billions of parameters) deployable on memory-constrained devices.

One of the strengths of this work is the detailed theoretical analysis that has been done in order to derive the error bounds on the relative quantization error. The authors have put commendable effort in stating the relevant preliminaries and building up to the theorems with appropriate lemmata which can be quite useful in general.

The paper is generally well-written.

**Weaknesses:**

I have some concerns in general, and would be happy to revise my score if these are satisfactorily addressed:

1. It is not clear what the novelty of this work is. It seemed to me that the core framework regarding the greedy quantization algorithm was already proposed in the prior work Lybrand & Saab (2021). In terms of proposed algorithm, is the only contribution of this work the replacement of the deterministic quantizer in Lybrand & Saab (2021) by a stochastic quantizer in order to make the analysis feasible for deeper neural networks? If so, the authors should clarify the fact that stochastic quantizer is NOT actually proposed first in this paper, since I could not find any relevant references to the same where stochastic quantization is introduced. Eq. (6) in this work is a well-known quantizer design. See for example:

(a) Non-subtractive dither (https://ieeexplore.ieee.org/document/823976) -- This is exactly the same, or for instance, see

(b) Eq. (4) of (https://ieeexplore.ieee.org/abstract/document/4542554)

The idea may date even earlier, but stochastic quantization is quite widely used for quantization in distributed optimization, etc. Keeping this in mind, it seems like the idea of replacing the deterministic quantizer of GPFQ with a stochastic quantization is straightforward. Although, the analysis of this and the conclusions is not trivial.

2. It is very important to keep in mind that stochastic quantization cannot actually be implemented in hardware as it is not easy to quantize to values that are not allowed by current hardware primitives provided by the GPUs (eg., 4-bit Floating point). This is important to noted because although the theory is pretty comprehensive with stochastic quantization, modifications must be made to the algorithm to account for hardware limitations. For instance, the stochastic quantizer ensures that the quantizer out put is unbiased, and this gives a nice upper bound on the expected quantization error. However, since this problem highly inspired from practical purposes, it is important to show some numerical simulation results for the non-deterministic version, especially since it might be likely that the bias in quantization errors from deterministic quantizers may propagate and accumulate in the greedy algorithm. For example, parallel research directions such as this: https://github.com/TimDettmers/bitsandbytes also work on neural network quantization, and assume that the weights are Gaussian, but translate to actual hardware benefits when deployed.

3. The work only considers fully connected neural networks and subsequently, uses results from vector quantization. In order neural network models like attention, parameters can be matrices instead, where low rank compression becomes important in addition to quantization. It is important that the authors highlight this limitations.

4. Please move the numerical simulations to the main paper. A problem which is inspired by a practical issue as this, should be justified with numerical simulations. Also, most of the times, GPFQ performs better than SPFQ. Why should SPFQ be preferred in that case? Expect for the theoretical guarantees?

5. Since the weights in this greedy algorithm are quantized sequentially, how is this sequence actually determined? Different weights in a networks have different sensitivities to the loss function -- is that taken into account while determining the sequence of quantization?

6. Please add a conclusions section.

**Questions:**

In addition to the concerns mentioned in the weakness section above, I have a few more questions. I'd really appreciate it if the authros could address them:

1. Comparing the assumptions in Thm 3.1 (infinite quantization points) and Thm. 4.1 (finite quantization points), it looks like the Gaussian weight assumption is only used in the latter. If I understand correctly (please correct me if I'm mistaken), then this assumption allows from a finite bit requirement because of strong concentration properties of the Gaussian distribution. Analysis of the expected quantization error for the Gaussian weights assumption yields a bit-requirement of log log N per parameter.

A similar result can be obtained without the Gaussian weight assumption in the worst-case analysis. This is done by side-stepping the Gaussian assumption by utilizing a Randomized Hadamard transform (which has similar concentration properties that can be obtained using Chernoff bounds). This was done in this recent work (https://ieeexplore.ieee.org/abstract/document/10095529) for linear regression and classification models. See Table 1. Since the error bounds essentially depend of the vector quantization error for Lipschitz activations, can similar results be derived utilizing the Hadamard transform based randomized quantizer as proposed in this work to sidestep the Gaussian weights assumption ,and without changing the final conclusion of the results?

I would be more than happy to revise my score post-rebuttal if I am mistaken anywhere and/or if the concerns are adequately addressed.

---

> ### Author Response · Authors · 2023-11-21
>
> We thank the reviewer for identifying the theoretical analysis as the strength of this paper. We agree that obtaining the first theoretical guarantees on the quantization error associated with deep neural networks is a valuable contribution.  We also believe that the fact these guarantees are obtained for a computationally efficient algorithm that can be used in practice adds to the value of the paper. Below are our responses to your comments and questions:
>
> **Response to weakness 1** We thank the reviewer for this comment. To be clear, nowhere do we claim that we are the first people in the history of quantization to propose stochastic quantization, or randomized rounding. As the reviewer noted, this has a long history in electrical engineering. That said, our contribution is to employ the stochastic quantizer in order to prove the first full-network theoretical error bounds, as previously noted. We have added a sentence (highlighted in blue) to this effect, see bottom of page 3.
>
> **Response to weakness 2:** We again thank the reviewer for the opportunity to clarify a misconception. First, we note that the mapping of weights to  alphabet elements  (i.e., weight quantization) is not intended to be done in hardware but is in fact expected to be done on a computer after the neural network is trained. The quantized network can then be built into hardware. This differs from signal processing applications where measurements are taken and quantized in real-time. That said, in the context of neural networks, the quantities that need to be "quantized" in hardware in real-time are the activations, but this is done deterministically, via simple bit-shift operations and is not the subject of our paper. Nevertheless, we stress that our analysis has no trouble with such quantization of activation functions. Activation quantization (in contrast with weight quantization) mathematically amounts to rounding the activation to its nearest element in the alphabet, and this rounding is a Lipschitz function hence all our theory applies.
>
> **Response to weakness 3:** This is not a limitation of the algorithm, though we did not theoretically analyze self-attention. To see that the algorithm can be readily applied to attention layers, note that these are of the form $\rho(XW_Q W_K^T X^T)X W_V$. As such, one can simply use SPFQ to quantize each column $w$, of $W_{K,Q,V}$, such that $XW \approx XQ$. If instead, a low-rank approximation is employed, so that $W=LR$, one can first quantize $L$ using SPFQ, then quantize $R$ as if it were the weights of the next layer.
>
> **Response to weakness 4:** Thank you for this comment. We believe that theoretical work is of value in machine learning. As such, we focused our attention in this paper on proving that a variant of GPFQ, namely SPFQ, can provide full-network error bounds. We provided a short numerical experiments section to validate that the method is viable in practice,
> and performs comparably to GPFQ (which was extensively compared to other methods in the original GPFQ paper: J. Zhang, Y. Zhou, R. Saab, "Post-training quantization for neural networks with provable guarantees", SIMODS, 2023). Indeed, the numerics suggest that SPFQ can outperform GPFQ in some cases. Due to its stochasticity, SPFQ if run multiple times, can potentially even provide significantly improved performance at the cost of multiple runs. While we appreciate the practical importance of such explorations, we note that in order to explore these issues one would need to run extensive numerical experiments (as done in the GPFQ paper). Given the purpose of this paper, and the computational resources needed for running such experiments, we choose to keep the focus on theory.
>
> **Response to weakness 5:**  This is a good question. At the moment, we use the built-in order. That is, the first entry is quantized first, and so on. However, there may be value in identifying better orderings, which we leave to future work.
>
> **Response to question 1:** Thank you for giving us the opportunity to clarify this misconception. The Gaussian assumption is NOT used in theorem 4.1, which holds deterministically. Gaussianity is only assumed in Theorem 4.2, simply to illustrate how Theorem 4.1 can also be used to obtain relative (vs absolute) error bounds. Here, the reviewer's intuition is correct. The randomness of the weights allows for the denominator to concentrate in norm.
>
> **Response to question 2 ("Since the error bounds essentially...conclusion of the results?"):**  This is a good question. The idea of using a randomized Hadamard transform (which is standard in compressed sensing literature, and the literature on fast Johnson-Lindenstrauss transforms, and in randomized numerical linear algebra) is often a reasonable one. However, in this context, it is not clear how it applies since it would require transforming the weights which would in turn require transforming the inputs to each layer in order to preserve the output.

---

> > ### Comment · Reviewer_sBRp · 2023-11-23
> > **Response and acknowledgment**
> >
> > Thank you for your response. Also, thanks for mentioning prior work for dithered quantization.
> >
> > I have a philosophical difference in the importance of theoretical analyses in practically inspired problems. NN quantization is a highly practically motivated problem, and one would expect numerical simulations to substantiate the theory (as often is a concern raised by other reviewers). I still believe the theoretical contributions of this paper is quite good, but unfortunately, I cannot advocate for the numerical section.
> >
> > Furthermore, in hardware, stochastic quantization is not easy to implement -- hence dithering is often a simplifying assumption for analysis. So, it is OK to use it for theory, but the fact that deterministic quantization performs at par with stochastic quantization may not be surprising, and consequently, the idea proposed in the proposed in the original GPFQ paper might be the go-to choice for practitioners.
> >
> > Next, the primary motivation for NN quantization is to deploy it on a memory-constrained hardware. I am not even talking about quantizing activations, but only the parameter weights. I find issues with the authors' sentence: *we note that the mapping of weights to alphabet elements (i.e., weight quantization) is not intended to be done in hardware but is in fact expected to be done on a computer after the neural network is trained. The quantized network can then be built into hardware.* -- How do you build a quantized network into hardware if your alphabet elements are represented in full precision? It **does not** represent a reduction in memory constraint. Quantizing the alphabet elements using low-precision hardware might introduce additional errors which is not taken into account by the analysis.
> >
> > It is not really clear how the algorithm can be readily applied to self attention. There has been a lot of work on low-rank approximation of weight matrices. It'd be appreciated if the authors can either elaborate more on it, or just address it as a limitations / future direction.
> >
> > To summarize, I do believe the theoretical contribution of this work is good. But there is scope to improve the paper.

---

> > > ### Author Response · Authors · 2023-11-23
> > > **Clarification**
> > >
> > > Stochastic quantization is a process that is not directly executed by hardware. Rather, it's a method used to derive quantized weights, which are values of limited precision. It is these quantized weights that are actually used in hardware.
> > >
> > > Separately, the procedure that converts high-precision weights into these quantized weights is carried out on a computer. This involves a stochastic quantizer. After this process is completed and the (finite precision) quantized weights are determined, they can be applied in hardware devices. To summarize, an algorithm is run on a computer to produce these limited-precision weights, and then these weights are utilized in hardware systems.

---

### Meta-Review · Area_Chair_YsdP · 2023-12-06

**Metareview:**

This paper proposes a quantization algorithm for compressing pre-trained neural networks. Although the authors tried to answer the reviewers’ questions, the reviewers are still not convinced with rebuttal and do not support the publication of the paper. Main concern is about the numerical experiments of the paper which are not sufficient to show the efficiency of the method. Please incorporate the reviewers’ comments and suggestions to improve the paper for the future submission. The authors could consider to submit their work to some journal.

**Justification For Why Not Higher Score:**

The reviewers do not support the publication of the paper.

**Justification For Why Not Lower Score:**

N/A

---

### Decision · Program_Chairs · 2024-01-16

Reject